

# Societal breakdown as an emergent property of large-scale behavioural models of land use change

Calum Brown[1], Bumsuk Seo[1], Mark Rounsevell[1,2]

[1]Institute of Meteorology and Climate Research, Atmospheric Environmental Research (IMK-IFU), Karlsruhe Institute of
5 Technology, Kreuzeckbahnstraße 19, 82467 Garmisch-Partenkirchen, Germany
[2]School of Geosciences, University of Edinburgh, Edinburgh EH8 9XP, UK

*Correspondence to*: Calum Brown (calum.brown@kit.edu)

**Abstract.** Human land use has placed enormous pressure on natural resources and ecosystems worldwide, and may even prompt socio-ecological collapses under some circumstances. Efforts to avoid such collapses are hampered by a lack of
10 knowledge about when they may occur and how they may be prevented. Computational models that illuminate potential future developments in the land system are invaluable tools in this context. While such models are widely used to project biophysical changes, they are currently less able to explore the social dynamics that will be key aspects of future global change. As a result, strategies for navigating a hazardous future may suffer from 'blind spots' at which individual, social and political behaviours divert the land system away from predicted pathways.

We apply *CRAFTY-EU*, an agent-based model of the European land system, in order to investigate the effects of human-behavioural aspects of land management at the continental-scale. We explore a range of potential futures using climatic and socio-economic scenarios, and present a coherent set of cross-sectoral projections without imposed equilibria or optimisation. These projections include various behavioural responses to scenarios including non-economic motivations, aversion to change, and heterogeneity in decision-making. We find that social factors and behavioural responses have dramatic impacts on
simulated dynamics, and can contribute to a breakdown of the land system's essential functions in which shortfalls in food production of up to 56% emerge. These impacts are largely distinct from, and at least as large as, those of projected climatic change. We conclude that the socio-economic aspects of future scenarios require far more detailed and varied treatment. In particular, the extent of economic 'irrationality' at individual and aggregate scales may determine the nature of land system development, with established pathways being highly vulnerable to deviation from this theoretical optimum.

## 25  1 Introduction

Human use of land resources has led to transformation of much of the Earth's surface (Hooke and Martín-Duque, 2012; Pongratz et al., 2008; Ramankutty et al., 2008). This transformation has enabled rapid rises in human population sizes and some living standards, but has also been a driving force of climate change and mass extinction (Newbold et al., 2016; Steffen et al., 2015). These consequences have become so severe that they threaten the continued provision of many of the essential
'contributions to people' that terrestrial environments make (Díaz et al., 2018). Societies now face the enormous challenge of



sustaining these contributions while simultaneously overcoming ingrained inequalities in their distribution (United Nations, 2017).

Computational models play a crucial role in understanding global change and identifying strategies to avoid its worst impacts. However, the systemic complexity that makes these models essential also makes them difficult to verify, inevitably incomplete

and therefore of limited accuracy (Beven, 2007; Brown et al., 2016a; Smith, 2001). Indeed, recent research suggests that land system models tend to produce unrealistic and inconsistent projections of human behaviour in particular (Alexander et al., 2017; Brown et al., 2019; Searchinger et al., 2017; Turner et al., 2018). This may make these models inapplicable in exactly the circumstances where they are most required; when socio-ecological processes break down and systems collapse (Cumming and Peterson, 2017).

One necessary improvement in modelling practice is the adoption of a wider range of conceptual and technical approaches (Alexander et al., 2017; Huber et al., 2018; Meyfroidt et al., 2018). At present, a small number of simplifying assumptions have become standard in land system modelling, allowing models to operate over large geographical extents and thematic areas without becoming computationally intractable. Broad assumptions about human behaviour are particularly common, usually following a paradigmatic reductionist approach that emphasises the role of macro-economic drivers of land use change

(Brown et al., 2016a, 2017; Calvin and Bond-Lamberty, 2018). These assumptions tightly constrain the representation of human decision-making, often forcing it to adhere to exogenously imposed equilibria. Furthermore, a focus on the agricultural sector has meant that other sectors (e.g. forestry, urban development) have generally been treated as separate systems rather than interacting components of the land system as a whole (Brown et al., 2017; Smith et al., 2010; van Vliet et al., 2019).

These shortcomings particularly constrain exploration of the effects of the social aspects of future scenarios, which, while

often quite dramatic, are not reproducible through the predominantly biophysical parameters of most land use models (Müller-Hansen et al., 2017; Riahi et al., 2017). Alternative, well-supported conceptualisations of the human land use system are available, and some have been formalised in agent-based or behavioural models that focus on individual-level decisions from which system properties emerge (e.g. (Arneth et al., 2014; Brown et al., 2016b, 2017; Fagiolo and Roventini, 2017; Magliocca, 2015; Rounsevell et al., 2012b)). To date, these models have been limited in scope, mainly operating only in specific contexts

or over small geographical areas (e.g. (An, 2012; Brown et al., 2017; Robinson et al., 2018)). However, their focus on underlying processes makes them suitable for scaling out and scaling up, across entire, coherent land systems (Rounsevell et al., 2012a). Recent conceptual and technical developments make this scaling feasible (Arneth et al., 2014; Verburg et al., 2015), and associated studies suggest that micro-scale behavioural processes can have significant macro-scale effects (Bai et al., 2016; Blanco et al., 2017a; Brown et al., 2018b; Calvin and Bond-Lamberty, 2018).

If a new generation of behavioural models are to make a substantial contribution to Earth System modelling, they must satisfy a number of requirements. First and foremost, they must achieve accuracy in their representation of basic processes that transcend land sectors, geographical areas and scenario conditions. Given this, models can move beyond context-specific pattern matching and retain sufficient flexibility to explore land system development under uncertain future global change. By the same token, these models need to incorporate relevant decision-making processes at a range of scales, from individual to





community and government, so minimising the role of exogenous and potentially inconsistent assumptions about nested actions (Galaz et al., 2012; Lippe et al., 2019; Rounsevell et al., 2014). Beyond behaviour, models must also reflect the true range of land use options, including gradients from subsistence production to profit-maximisation, highly extensive to highly intensive management, and entirely uni-functional (monocultural) to highly multifunctional or mosaic land systems

(McDermid et al., 2017; Verburg et al., 2012).

In order to move towards these goals, we have developed CRAFTY-EU, a continental-scale, agent-based model of the European land system based on the CRAFTY modelling framework (Murray-Rust et al., 2014). We describe the design, calibration and evaluation of this model before using it to explore future developments in Europe's land system under a range of climatic and socio-economic scenarios. We assess the sensitivities of these developments to scenario conditions and various

forms of land manager behaviours, and their implications for the supply of a range of ecosystem services and land system stability. We then discuss the possible impacts of human behaviour within the land system, as well as the value of novel modelling approaches of this kind for understanding and managing Earth System change.

## 2 Methods

CRAFTY-EU is an application of the CRAFTY framework for agent-based modelling of land use change (Blanco et al., 2017a;

Brown et al., 2018b; Holzhauer et al., 2019; Murray-Rust et al., 2014). The CRAFTY framework allows land use outcomes to be modelled as the result of decision-making and competition among individual agents, each of which can represent an individual or multiple land managers, and which produce a range of ecosystem services. Production levels are determined by the productivity of the land (defined through a range of natural and anthropogenic capitals, as described below), the intensity of land management, and agents' willingness or ability to produce certain ecosystem services. Agents are grouped into Agent

Functional Types (AFTs) (Arneth et al., 2014) on the basis of their management intensity and decision-making characteristics, such as degree of focus on profit-generation and desire to maintain an existing land use. Variation within AFTs allows for individual differences in production levels and land management decisions. Therefore, the model allows for emergent land system properties that are not constrained by assumptions about optimality, equilibrium or economic rationality. The main components of the applied model are summarised in dedicated sections below.

CRAFTY-EU is calibrated using outputs from the IMPRESSIONS Integrated Assessment Platform (IAP), a cross-sectoral, multi-model tool for simulating European land system change (Harrison et al., 2015, 2019; Holman et al., 2017). All necessary input data (described below) are derived from this source, ensuring the transparency and internal consistency of the implementation. This model pairing also allows socio-economic and climatic scenarios to be defined on the basis of comprehensive, cross-sectoral simulations of the European land system that have been extensively evaluated, validated and

utilised (Brown et al., 2014a; Harrison et al., 2012, 2016, 2019; Kebede et al., 2015; Pedde et al., 2019b). Changes in the modelled land system are therefore attributable either to CRAFTY model dynamics (investigated below) or scenario



conditions, rather than internal inconsistencies in input data from different sources. Full details of the calibration of CRAFTY-EU are given in Appendix A.

## 2.1 European application

CRAFTY-EU covers the European Union-27 (EU member states that include the UK, but exclude Croatia) together with Norway and Switzerland. The model operates at a 10' (arcminute) resolution, with 23,871 grid cells in total. This resolution was selected for its consistency with input data, all of which had the same resolution, for its low computational demands, allowing multiple model runs to be carried out quickly, and because of a shortage of appropriate calibration data at finer resolutions. Nevertheless, this resolution is relatively coarse for an agent-based model application, and means that modelled agents cannot be seen as representative of individual real-world land managers in most cases. Instead, they are drawn from semi-aggregated AFTs designed to represent coherent localised land use systems (Letourneau et al., 2012), with management and behavioural characteristics expressed at appropriate generality, as described below.

## 2.2 Agent Functional Types

Agent Functional Types used in CRAFTY-EU were designed to provide generic coverage of the major sectoral and cross-sectoral land systems at local (10') scale across Europe. Key distinctions were made between levels of management intensity and between the ranges of ecosystem services produced (Arneth et al., 2014; Letourneau et al., 2012; Murray-Rust et al., 2014; Paul et al., 2017). The final typology was intended to capture the primary form of land management within each grid cell, while allowing for secondary land uses and variation in local land management practices (Table 1; Appendix A). The initial distribution of these AFTs across the modelled land surface was based on the distribution of land use categories modelled by the IAP under baseline conditions (Appendix A), ensuring consistency across initial simulation conditions, and comparability with subsequent scenario-based changes.

The abilities of different AFTs to utilise capitals and produce ecosystem services were defined via capital sensitivity and productive ability parameters (summarised in Table 1 with further details and exact parameterisations in Appendix A; see also Murray-Rust et al. 2014). Where possible, values were derived from simulated production data in the IAP, and otherwise assumed on the basis of land management intensity and diversity. Behavioural differences between AFTs (in terms of willingness to change land use or abandon land, and range of variations in capital sensitivities and ecosystem service production levels) were also introduced to assess the robustness of model outcomes to behavioural variations (see below and Appendix A). Urban land use was not actively modelled, but constrained to follow the results of the IAP, which includes advanced modelling of urban development (Terama et al., 2019).



## 2.3 Land productivities (capitals)

The productive potential of each modelled grid cell was described via five capitals: natural capital (crop productivity, grassland productivity, forest productivity), human capital, social capital, manufactured capital and financial capital. Each capital was derived from the IAP as described in Appendix A. Scenario-specific changes in capital values were produced by running the

IAP under each scenario in turn and repeating the derivation process. Each of the productivity capitals accounts for climate-induced changes during the period of simulation, including effects of changes in temperature, precipitation and $CO_2$ levels. These changes were simulated for the IAP by combinations of global and regional climate models: EC_Earth/RCA4 for RCP2.6, and HADGEM2-ES/RCA4 for RCPs 4.5 and 8.5 (Harrison et al., 2019). Socio-economic conditions (as defined by the SSPs; (Riahi et al., 2017)) affected anthropogenic capitals (human, social, manufactured and financial) as determined via

a stakeholder-led elaboration of scenario narratives and a subsequent uncertainty-based quantification (Harrison et al., 2019; Pedde et al., 2019b). Because IAP outputs were only available at three timeslices (2020s, 2050s and 2080s), capital values were linearly interpolated to give annual values for each grid cell over the period 2010-2086.

## 2.4 Ecosystem services, demand and supply

The CRAFTY framework is designed to account for the demand and supply of a range of ecosystem services, and we incorporate a representative group for which calibration data are available or for which assumptions related to calibrated land management can be made: timber, meat, crops, carbon sequestration, landscape diversity and recreation. Annual demand levels for each of these services were derived from IAP outputs, via conversion of simulated land cover to service production levels as described in Appendix A. All demand levels are available in Supplement 1.

Demand levels were converted to 'benefit' values that describe the relative benefit gained for production of each service by simulated agents. These values were calculated for each agent and each cell, and used as a basis for competition for land between agents, with the agents producing the most (or the most highly valued) services gaining the highest benefit values and therefore best-placed to win the competition for cells (Appendix A) (Holzhauer et al., 2019; Murray-Rust et al., 2014). However, 'rational' competition was not enforced, meaning that agents with the highest benefit values were not necessarily

allocated land, depending on decision-making parameters (outlined below). In addition, benefit values were not designed to ensure full supply of each service, but only to respond in defined ways to changes in demand and supply levels, stimulating production, but not guaranteeing a given production level. The model therefore contains no assumptions that override the emergence of sub-optimal or non-equilibrium outcomes from scenario conditions.

Another key feature was that demand levels were normalised to produce the same benefit for supply of each proportional unit

of unmet demand. This means that production was assigned the same value at any given level of unmet demand for each service. Service production in any part of the EU contributed to satisfying demand levels, representing an assumption of free trade across the modelled area (constrained by the infrastructure and transportation networks described in the manufactured



capital values). This is a reasonable assumption given that the EU is a free trade zone. A full description of the valuation and competition process is given in Appendix A.

## 2.5 Model evaluation

The CRAFTY modelling framework has been extensively evaluated and applied in previous studies (e.g. (Alexander et al.,
2017; Blanco et al., 2017a; Brown et al., 2014b, 2018b; Holzhauer et al., 2019; Murray-Rust et al., 2014)), as has the IAP upon which this application of CRAFTY is based (e.g. (Brown et al., 2014a; Harrison et al., 2016; Holman et al., 2017; Kebede et al., 2015)). Both sets of evaluation have included sensitivity and uncertainty analyses (Brown et al., 2014a, 2014b, 2018b; Kebede et al., 2015; Synes et al., 2018), comparisons to empirical data and to the results of other models (Alexander et al., 2017; Blanco et al., 2017a), full descriptions of model design and functioning (Harrison et al., 2015; Murray-Rust et al., 2014)
and full, free access to the models themselves including interactive online systems for exploring model outputs (Holzhauer et al., 2016; IMPRESSIONS Project, 2018); https://landchange.earth/CRAFTY). Both models have also been extensively used in, and informed by, a stakeholder engagement process that has occurred over several years across the EU (Kok et al., 2018). Here, additional model evaluation focused on the behaviour, stability and interpretability of the European application of CRAFTY. These characteristics were primarily assessed through two sets of runs under static, baseline conditions, starting
from an unassigned (empty) land use map and from the baseline land use map derived from the IAP. The purpose of these two exercises was, respectively: 1) to check whether baseline conditions would generate a 'realistic' land use configuration purely on the basis of capital levels and AFT characteristics (i.e. in the absence of any spatial information about land management), and 2) to check for divergence in outcomes from a common starting point consistent with the starting point of other scenario runs. Model dynamics were checked visually and statistically, using the numbers of agents within each AFT and levels of
service provision. Both evaluation exercises are described in detail in Appendix B.

## 2.6 Simulation schedule

CRAFTY-EU runs on annual timesteps at which a proportion of cells are subject to potential abandonment, adoption, or competition (Murray-Rust et al., 2014). In the first evaluation exercise, the model was run over 800 timesteps, with 20% of cells being randomly selected for potential change (i.e. the maximum number of cells that could change at each time step, if
required by the competition process). This arbitrary but high rate of competition allowed for rapid changes to the simulated land system, ensuring model dynamics could be clearly perceived. The period required for the model to reach a steady-state was identified, and 10 further independent simulations were then run to this point using different random number generator seed values. The second evaluation exercise was performed over 100 timesteps, again with a 20% rate of cell selection. This exercise was designed to run a sufficient number of replicates to identify and understand any divergence from stationarity in
model dynamics.

Following the evaluation exercises, simulations were run for 71 timesteps, representing the period 2016-2086, with 5% of cells selected for potential change at each of these timesteps. As an upper limit, this rate is up to an order of magnitude greater than





observed (e.g.(Loveland et al., 2012)) and projected land use changes (e.g.(Schmitz et al., 2014)), allowing for the majority of potential changes to be rejected while maintaining scope for rapid land use change under extreme scenarios. These simulations all began from the baseline land use map (Fig. A1.2), and proceeded according to scenario conditions in terms of ecosystem service demand levels and capital values (Appendices 4 and 5). Seven distinct scenarios were simulated, each of which

comprised a combination of Representative Concentration Pathways (RCPs) and Shared Socioeconomic Pathways (SSPs) (O'Neill et al., 2017) as described in Table 2. Socio-economic scenarios were developed from the SSPs through a stakeholder-engagement process described in detail in Kok et al. (2018). These scenarios were first run through the IAP as described above in order to produce representative levels of capitals and demands for use in CRAFTY-EU. Throughout each simulation, land use maps, numbers of agents in each AFT, ecosystem service production levels and fragmentation indices (fractal dimensions)

were recorded.

For each scenario, five distinct parameter sets were applied to assess the effects of variations in agent's modelled behaviours (full parameterisations and explanations for each of these are given in Appendix A). These parameter sets differed in terms of the abilities of agents to produce services and their tolerance of low benefit values and of competition, all of which varied at AFT and individual agent levels. These variations were designed to represent general behavioural effects arising from land

managers' decision-making, accounting for aggregation to the model's spatial resolution. Under the 'baseline' behaviours, agents persisted with land uses that provided benefits unless outcompeted by other agents, and did not vary at individual level. In other parameter sets ('increased thresholds') agents were less tolerant of low benefit values and competition (i.e. required larger returns to continue their land management or to switch to another) and varied individually in their tolerances and service production levels ('individual variation'). In each case, relatively small and large deviations from the baseline parameter values

were used.

## 3 Results

### 3.1 Model evaluation

#### 3.1.1 Simulations with no initial land use map

Simulations initiated under all baseline conditions, but without the initial land use map, were found to quickly converge to an

approximate steady-state (Fig. B1), but not to achieve formal stationarity over 800 timesteps (Box-Ljung test p-values <0.01 for numbers of agents belonging to each AFT and service production levels over 50-timestep periods). This appeared to be due to path-dependent oscillations (over short- and long-timespans) that, while statistically significant, were small relative to total agent numbers and rarely affected the relative rank of each AFT (Fig. B1). These oscillations were amplified by the high rate of competition for cells allowed in the evaluation simulations (20% of cells at each timestep), and as such remained broadly

in line with expectations, with no evidence of either ongoing systematic change or dramatic regime shifts.



The 300[th] timestep was chosen as representative of model outcomes following the initial period of rapid change, and the numbers of agents belonging to each AFT at this point in each of the ten independent simulations were then plotted (Fig. B2a) along with the proportional supply levels of each service (Fig. B2b). These results showed strong convergence between simulation outcomes, with both relative and (approximate) absolute numbers of agents being reproduced in each simulation. Service levels remained between 95% and 110% of demand levels in all cases. In these relatively unconstrained circumstances, the model tended to produce a slight excess of meat and carbon sequestration services, with a predominance of multifunctional AFTs and a relative lack of intensive-management AFTs. However, aggregated AFTs showed not only spatial consistency across the simulations, but also agreement with the (unutilised) baseline map (Figure B3), suggesting that the model spontaneously produced realistic land use configurations on the basis of land productivities, AFT parameterisations and demand levels.

### 3.1.2 Simulations from baseline map

The simulation initialised with the baseline land use map and run under static conditions remained stationary (Box-Ljung test p-values >0.1 for numbers of agents belonging to each AFT and service production levels) (Fig. B4). The total number of agents within each AFT barely changed, with the maximum range in number of agents over the course of the simulation being 2. Further realisations were not generated given this lack of variation and the model stability that it demonstrated under static conditions.

### 3.2 Scenario simulations

Scenario simulations showed widely divergent land systems being produced by the mid-2080s under different scenario combinations, which were not substantially reduced by behavioural variations between agents (Table 2, Supplement 2). These differences were primarily driven by socio-economic scenario conditions, but also by different levels of climate change between the three climate scenarios used (Figs. 1 & 2). Broadly, where socio-economic capital levels were maintained or increased, the land system diverged from the baseline scenario by a relatively limited amount, with widespread intensive management of land and small shortfalls or surpluses of most modelled services. Conversely, where these capitals declined substantially, widespread extensification and abandonment of land occurred and large shortfalls in service levels developed (Fig. 1, Table 2, Supplement 2). These dynamics were partly ameliorated by increases in productivity in some areas associated with high-end climate change, particularly north-western Europe.

Of particular importance were manufactured and financial capitals, which increase greatly (up to 250%) in some scenarios (e.g. SSP1) and decrease (by around 90%) in others (e.g. SSP3), depending on scenario storylines (Fig. 1 & Table 2). These capitals are crucial in supporting intensive land management in *CRAFTY-EU* (Appendix A), and so determine the scope for the most productive uses of land. Where these capitals increased, surpluses of services (especially food) developed, and where they decreased, shortfalls developed, reaching 56% of food demand in the RCP4.5-SSP3 scenario combination (Fig. 2).



We simulated three socio-economic scenarios in different climate scenarios, and all showed notable similarities between climates. SSP1 had the most consistent differences; this scenario has high demands for all services, and the difference between climate scenarios was due to increases in average crop and forest productivity capitals under RCP4.5 relative to RCP2.6. These productivity changes increased the competitiveness of intensive management enough to allow it to outcompete more extensive, multifunctional land uses, and so allowed production to increase enough to satisfy demand.

The most consistent and most negative scenario was SSP3, in which economic and social challenges led to disintegration of the land system across much of Europe, with large areas being abandoned, managed extensively, or fluctuating over time (Figure 1, Table 2, Supplement 2). These dynamics were particularly pronounced in more fertile areas of Europe, where currently dominant intensive management declined dramatically during the first half of the century. Similar results were found in both RCP4.5 and RCP8.5, suggesting that changes in climate were minor in comparison to the almost complete loss of financial and manufactured capitals that undermines the productive use of land in SSP3. Nevertheless, supply levels increased markedly towards the end of the century in RCP8.5, as increased natural capitals (i.e. yield increases) offset some of the losses from declining socio-economic capitals. Conversely, in technologically advanced scenarios (e.g. SSP4), where manufactured and financial capitals increase greatly, demands for services could be met relatively easily, leading to a decline in intensive management because of a lack of need, rather than a lack of opportunity.

Results also show some broad geographical patterns. While the most unproductive areas of Europe (e.g. mountain ranges, high latitudes) were the most resistant to change under any scenario, other areas responded differently depending on the scenario conditions. South-eastern Europe (Greece, Bulgaria, Romania and Hungary), was slightly more vulnerable to extensification and abandonment where supply levels matched demands as in SSP4 (i.e. when 'benefit' or profit levels were low), but were more robust to low levels of capitals in SSP3. In contrast, Western Europe (particularly Germany, France, England and intensively-managed areas of Spain) suffered widespread abandonment in SSP3. As climate change increased in magnitude through RCPs 2.6, 4.5 and 8.5, land management in North-Eastern Europe (Poland, Lithuania, Latvia, Estonia and southern Finland) tended towards forestry, as increases in forest productivity and decreases in crop productivity made arable agriculture less competitive.

Behavioural parameter variations had distinct effects in different scenarios and on different metrics (Table 2 & Supplement 2). In general, scenarios with less intensive management (and also lower land use fragmentation as measured by the fractal dimension) were less affected by behavioural parameter changes; these scenarios included RCP4.5-SSP3, RCP8.5-SSP3 and RCP4.5-SSP4. Conversely, scenarios with more intensive management (RCP4.5-SSP1, RCP2.6-SSP1 and RCP2.6-SSP4) were more affected, producing more fragmented land systems, but not necessarily different levels of ecosystem service supplies (Table 2). These differences were correlated with climatic scenarios, with the more productive land systems under high-end scenarios proving more robust to behavioural differences. Of the two forms of variation simulated, increased requirements for benefit from land management (thresholds) led to increased fragmentation within scenarios on average, but also increased differences in fragmentation between them. Individual variation increased the differences in fragmentation more than the



average, but at the higher strength these differences were reduced to approximately baseline levels, suggesting a 'peak effect' under small levels of individual variation beyond which the most irrational agents were selected out.

In all cases, the delicate balance between food and timber production highlights the sensitivity of results to demand levels for ecosystem services derived from agriculture and from forestry. In many cases, simulations resulted in widespread adoption of

multifunctional land uses that provide both sets of services to some extent, with the locations of these being scenario-dependent. The levels of demand, relative valuation and production of these services therefore appear to be major determinants of the nature of European land systems in this model.

## 4 Discussion

The work presented here highlights the importance of both model design and scenario conditions for understanding possible

future change in large-scale land systems. This complements previous findings that model design and initial data conditions had a greater impact than scenarios on simulated land use change (Alexander et al., 2017), but extends the comparison to new design and scenario components. Until now, exploration of these has been generally limited to optimising pattern-based models and the biophysical and economic factors that they incorporate, neglecting the social conditions and processes that often vary dramatically between scenarios (Brown et al., 2017; von Lampe et al., 2014; Pedde et al., 2019a).

This model implementation demonstrates that agent-based modelling of socio-ecological systems at continental scales is both a feasible and informative method for scenario exploration, producing clear and distinct outcomes that respond directly to scenario definitions. These responses include breakdown of the simulated land system, in which rapid and sub-optimal land use changes lead to severe shortages of ecosystem services including food. While such breakdown is occasionally a feature of real-world land systems and a plausible result of severe pressures in the future (Ehrlich and Ehrlich, 2013; Hazell and Wood,

2008; Weiss and Bradley, 2001), it is largely beyond the reach of conventional modelling approaches (Balint et al., 2017; Brown et al., 2016a; Farmer and Geanakoplos, 2009). The ability to explore such breakdowns is clearly necessary for attempts to achieve the converse; stability and sustainability in socio-ecological systems.

To allow proper interpretation, the remainder of this Discussion is divided between technical considerations relating to model design and parameterisation, and reflection on the results produced in this study.

### 4.1 Model design

CRAFTY-EU is an explorative model, and is not designed to predict (inherently unpredictable) land system changes (Brown et al., 2016a). Further, the CRAFTY framework is intended to provide relatively simple, generic methods for exploring land manager decision-making over large geographical extents (Murray-Rust et al., 2014), and is used here to represent decision-making within local land systems rather than at the level of individual managers. As such, this model application is a first-step

towards improved understanding of behavioural processes within large-scale land systems.



At a general level, the results presented here are realisations of a single approach to land systems modelling, which complement alternative projections made by other models (e.g. (Harrison et al., 2019; Stürck et al., 2018; Verkerk et al., 2018)). In particular, conceptual or theoretical frameworks within which behavioural modelling can occur are diverse and disputed, and a universally applicable representation of the complex social processes involved in land use change is not available or even, necessarily, possible (Brown et al., 2016a; Huber et al., 2018; Meyfroidt et al., 2018). Even given this caveat, our exploration of behavioural parameters is illustrative rather than exhaustive, intended to reveal the implications of basic assumptions more than exact parameter values. Indeed, the CRAFTY framework is designed specifically to allow exploration of abstracted behaviours that do not require precise parameterisation. In this respect, this study deliberately builds on earlier studies of the parameterisation of behavioural processes in CRAFTY, including in a similar scenario context (e.g. (Brown et al., 2014b, 2018b)).

A number of more specific considerations are also important for interpreting our findings. Most significantly, the simulations presented here form an experiment into the effects of simulating land management as the provision of multiple (but arbitrarily limited) ecosystem services, which depend upon a set of scenario-dependent capitals and which are valued equally per standardised unit of demand. This design ensures that trade-offs between services are clear, but does not assume preferential production of some services (such as food) when supply levels are equally insufficient. As a result, scenarios in which shortfalls in service provision exist represent an artificially balanced outcome, with real-world equivalents expected to diverge towards more homogeneous land uses to some extent. In this respect, our findings suggest that further exploration of trade-offs between service provision, in terms of both production systems and valuation, should be a priority for land system modelling. This is especially important given potential changes in current valuation practices, for example through carbon pricing or payments for ecosystem services, which could transform the competitiveness of currently minor land uses and require models to account for the services that they produce (Kay et al., 2019).

Beyond Europe, neither CRAFTY-EU nor the IAP that is used to calibrate it explicitly represent production and trade. While scenario-specific import levels are assumed, these are likely to be overestimates in challenging scenarios with large shortfalls in service provision that imply shortages elsewhere in the world (Dellink et al., 2017; Harrison et al., 2015; Stevanović et al., 2016). Furthermore, alternative treatments of international trade based on assumptions of economic equilibrium would be inconsistent with the supra-economic behavioural approach used in CRAFTY-EU (Arthur, 2006). The relative provision of different services is also subject to substantial uncertainty in our representation of forest growth, with assumed adaptation to changes in species' suitability likely to overestimate real-world adaptation (Schelhaas et al., 2015), as the CRAFTY framework has previously been used to demonstrate (Blanco et al., 2017b, 2017a).

Notwithstanding the above limits on the model's accuracy, the robust, cross-sectoral nature of the model, building on the established and evaluated IAP and CRAFTY framework, means that it is capable of providing well-founded and novel insight into land system dynamics. Model evaluation performed for this and earlier studies has revealed no clear biases or instabilities, with CRAFTY-EU producing realistic outcomes in the absence of information about baseline land uses. The responses of the





model to the scenarios can be seen as coherent responses to a set of land system drivers that are fully interpretable in light of transparent model assumptions.

## 4.2 Model results

A key finding of this work is that the sensitivity of land use to social (as well as economic and climatic) conditions makes land
systems vulnerable to breakdown when these conditions worsen substantially. Such worsening is a key characteristic of some future scenarios (e.g. SSP3), but one that has generally only been explored through qualitative scenario descriptions (e.g. (Cradock-Henry et al., 2018; Kebede et al., 2018; Pedde et al., 2019a)). In SSP3, declines in socio-economic capitals are so precipitate and substantial that the resulting breakdown of the simulated land system is highly plausible, and proves almost impossible to avoid in our modelling, regardless of exact parameterisations. In other model projections of this scenario, similar
outcomes are avoided only by very large increases in food prices that compensate for relatively low crop yields and stimulate food production at the expense of forest cover (Doelman et al., 2018; Fujimori et al., 2017; Harrison et al., 2019; Hasegawa et al., 2015; Popp et al., 2017).

This implies that scenario modelling using economic equilibrium assumptions could prove misleading where scenario conditions place limits on price or production levels. Substantial declines in financial and manufactured capitals, for instance,
may effectively preclude the necessary economic stimuli or production responses to meet demand in SSP3. While this problem is starkly illustrated by non-equilibrium modelling such as that presented here, its knock-on effects on consumption, demand and supply (and wider socio-economic systems) are obscured by the pre-definition of those factors in scenario storylines. The CRAFTY-EU model therefore makes only one of two crucial connections, linking social conditions and supply levels through the capitals-production relationships without completing the link back to demand.  Most starkly, if insufficient food is produced
to maintain population levels, populations and subsequent demand would inevitably decrease – a fundamental feedback that remains absent from scenario modelling. Such internal inconsistencies in modelled scenarios are not limited to socio-economic systems; scenario assumptions about the magnitude of climate change are widely made without accompanying assumptions about implied land-based mitigation actions (Kriegler et al., 2017). For analyses of future scenarios to be dependable, all of these issues need to be addressed.

In addition to identifying very large negative impacts of some scenario combinations, we also find that these impacts differ widely across Europe. Some areas appear to face high likelihoods of substantial changes; for example we find that Eastern Europe is broadly more vulnerable to changes in demand levels (and hence 'benefit' or price levels) and Western Europe broadly more vulnerable to changes in capital levels. Many of the worst simulated outcomes have notable mirror-images in history, where land systems gradually became more intensive, homogeneous and efficient as financial, technological and social
capitals developed (e.g. (Petit and Lambin, 2002)). Projected declines of these capitals produce a return to fragmented, extensive production in our simulations; a reverse precedent that adds some credibility to model responses, while clearly not suggesting predictive accuracy. It is also notable that greater climatic change can actually ameliorate the worst outcomes in some cases (e.g. SSP3) because it allows higher yields in parts of Europe to offset losses and socio-economic difficulties





elsewhere. Similarly, technologically advanced scenarios (e.g. SSP4) allow relative ease of production and therefore free-up land, leading to some extensification and abandonment.

Within these broad findings, variations in behaviour can have substantial effects. These are more pronounced in low-end climate change scenarios that rely on slight competitive advantages of intensive land systems to meet service demand levels, and which are therefore sensitive even to slight economic irrationality in management decisions. The literature suggests that intensive farmers are more vulnerable to changing price levels (van Vliet et al., 2015), and this vulnerability is amplified here by a reliance on socially-mediated capitals that support farming. It is also notable that the behavioural effects we observe are similar at both simulated strengths of behaviour, suggesting that even small differences in land managers' responses to scenario conditions can have substantial consequences. Indeed, increasing behavioural differences may lead to the loss of more extreme agents from the system, giving a behavioural saturation effect that could limit the extent of irrationality in the real world. In any case, the evidence of widespread deviation from economically optimal behaviour amongst land managers, such as selection of economically inferior options for social reasons, or socially-mediated uptake that spans long time periods (Brown et al., 2018a, 2019; Sereke et al., 2016), justifies – if not necessitates – the incorporation of such behaviour in land systems models. We suggest that this is a pre-requisite for accurate assessments of future scenarios, and so for effective land management planning and policy-making.

## 5 Conclusions

The application of an agent-based model to simulate future European land use change suggests an important role for large-scale behavioural models of this kind. CRAFY-EU is developed here to investigate broad forms of human behaviour in the context of land management decision-making, and demonstrates that such behaviours can have multiple substantial effects in different scenario contexts. Furthermore, the most notable of these effects were linked to basic model assumptions rather than exact design or parameterisation choices. The inclusion of socio-economic aspects of future scenarios as active drivers of land use decision-making had impacts at least as large as simulated climate change, with behavioural effects further shaping trajectories within those scenarios. Competition between a cross-sectoral, multi-functional range of land uses highlighted the critical importance of the relative valuation of ecosystem services, and the ability of models to represent a relevant range of services. Most prominent, however, was the effect of allowing land use decisions to occur without enforced equilibria or optimisation. In scenarios with challenging socio-economic conditions, this led almost invariably to breakdown of the simulated land system, and severe shortages of food and other services. These effects were apparent even at low levels of behavioural complexity, and persisted across tested parameterisations. Indeed, we find some evidence that behavioural effects may be partially 'self-correcting', with more extreme behaviours being selected out by a competitive process. These findings show a clear need and scope to consider the role of human behaviour in shaping land system development. Although this task remains challenging, the data and tools to explore social dimensions of scenario space are developing rapidly, and appear capable of providing important new insights into the future development of large-scale land systems.



**Code and data availability**

The full model code and date are available for download and visualisation at https://landchange.earth/CRAFTY

**Appendices**

Appendix A: Model parameterisation

Appendix B: Model evaluation

**Supplements**

Supplement 1: Demand files, giving ecosystem service demand values for each scenario and year.

Supplement 2: Further graphical results summaries.

**Author contribution:** CB developed the model and drafted the manuscript; BS & CB performed sensitivity analyses and BS developed calibration routines and the web platform; MR advised on model development and interpretation, and all authors finalised the manuscript.

**Competing interests**

The authors declare that they have no conflict of interest

**Acknowledgements**

This research was supported by the EU Seventh Framework Programme project IMPRESSIONS (grant no. 603416) and the Helmholtz Association





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





| Agent functional type | Ecosystem services produced | Area covered (% cells in baseline) |
|---|---|---|
| Intensive arable farming |  | 12.6% |
| Intensive pastoral farming |  | 4.8% |
| Intensive agro-forestry mosaic |  | 10.8% |
| Intensive farming |  | 5.9% |
| Managed forestry |  | 15.0% |
| Mixed farming |  | 5.2% |
| Mixed pastoral farming |  | 1.9% |
| Mixed forest |  | 0.3% |
| Extensive pastoral farming |  | 0.9% |
| Extensive agro-forestry mosaic |  | 4.8% |
| Very extensive pastoral farming |  | 2.3% |
| Multifunctional |  | 18.3% |
| Minimal management |  | 6.5% |
| Unmanaged land |  | 9.7% |
| Unmanaged forest |  | 0.3% |
| Peri-urban |  | 0.7% |

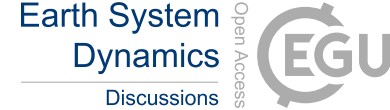

**Table 1: Details of Agent Functional Types (AFTs) used in *CRAFTY-EU*.** Ecosystem services are represented as follows: crops:

; meat: ; timber: ; carbon sequestration: ; diversity: ; recreation: . Primary ecosystem services of each AFT are those produced in quantities at least 50% of the maximum of any other AFT, and are shown in black. Secondary ecosystem services are those produced in lower quantities, and are shown in grey. The initial distribution of these AFTs across modelled grid cells, full parameterisation of capital sensitivities and production levels are described in full in Appendix A. The conceptualisation and parameterisation of AFTs allows for some variation in capital sensitivities, service production abilities and land uses within each AFT. The Urban (not shown) and Peri-urban AFTs are included only as placeholders for urban modelling in the IAP, and are constrained to reproduce the same results here, with Peri-urban also allowing for surrounding production of other ecosystem services as shown.





**Figure 1: Maps of simulated land cover in 2086 under the RCP4.5-SSP1 scenario combination (top left) and the RCP4.5-SSP3 scenario combination (top right), showing the two extremes of modelled outcomes across the simulated scenarios. These extremes are driven by the radically different socio-economic capital levels within the two scenarios (bottom; capitals shown as mean values, normalised by their initial mean value).**



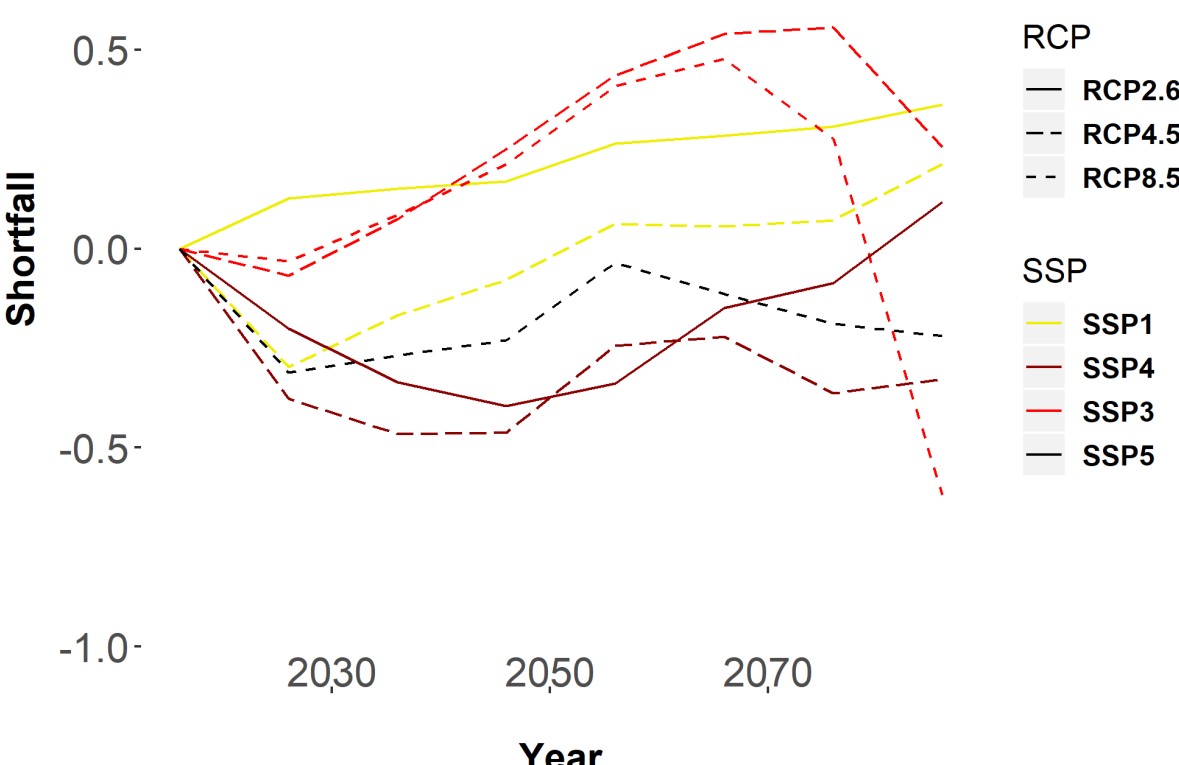

**Figure 2: Total shortfalls in food production (meat and crops) as proportion of demand levels in all simulated scenario combinations. Negative values indicate a surplus and positive values indicate a shortfall.**





| Scenario combination | Explanation | Main Results | Effects of behavioural variations |
|---|---|---|---|
| RCP2.6 – SSP1 | Represents a future in which limited climate change occurs, and socio-economic conditions gradually improve through economic growth, stable government, high social cohesion and international cooperation. | Gradually increasing shortfalls in supply levels of most services, especially timber, over the simulation period. Intensive management across much of Europe, with more extensive land uses in northern and southern latitudes. Relatively stable AFT dynamics. | Increased thresholds and individual variations produced more intensive & efficient land uses and more use of unmanaged land, but similar supply levels |
| RCP2.6 – SSP4 | Represents a future in which limited climate change occurs, but large economic inequalities and fluctuations develop and contribute to low social cohesion. Nevertheless, substantial technological investment is made and environmental protection is prioritised. | Broadly increasing service provision in first half of the century, driven by large increases in manufactured and financial capitals, leading to surpluses, especially of meat. Subsequent dramatic drops in intensively managed areas mid-century and tendency to abandonment, minimal management or extensive management, especially away from central Europe; development of shortfalls. Fragmentation of land use. | Increased thresholds and individual variations produced more intensive & efficient land uses particularly in Central-Western Europe, with substantial increase in meat supply and small drop in crops supply |
| RCP4.5 – SSP1 | Represents a future in which low-medium climate change occurs, and socio-economic conditions gradually improve through economic growth, stable government, high social cohesion and international cooperation. | Relatively stable service supplies but consistent shortfalls in timber production. Widespread intensive management of land, with little change from baseline. | Increased thresholds and individual variations produced more abandonment in Central-Eastern Europe, with more timber production and otherwise similar supply levels. |
| RCP4.5 – SSP3 | Represents a future in which low-medium climate change occurs, while social and economic conditions worsen, with limited and ineffective political responses. | A very dynamic scenario in which land uses fluctuate in response to rapidly declining capital levels. Very large shortfalls develop, especially of food, although these are rapidly reduced after 2070. Widespread extensification and abandonment of land occurs across Europe. | Very similar results across all parameterisations |
| RCP4.5 – SSP4 | Represents a future in which low-medium climate change occurs, and large economic inequalities and fluctuations develop and contribute to low | Substantial surpluses are produced thanks to increasing financial and manufactured capitals. Fluctuations in land management result in a | Similar results across all parameterisations, with behavioural differences leading to slightly less |



| | social cohesion. Nevertheless, substantial technological investment is made and environmental protection is prioritised. | changeable and fragmented land system, with extremes of intensive and very extensive land management co-existing in many areas. | extensification and slightly larger surpluses. |
|---|---|---|---|
| RCP8.5 – SSP3 | Represents a future in which high-end climate change occurs, while social and economic conditions worsen, with limited and ineffective political responses. | As with RCP4.5-SSP3, land management and service supplies are very dynamic, with different trajectories throughout the century, producing large shortfalls that are eventually overturned. Slightly increased average crop productivity supports some intensive management in an otherwise highly fragmented, extensively-managed land system. | More intensive management in Central-Western Europe and more abandonment in Eastern Europe, giving similar service levels with larger surpluses by the end of the period. |
| RCP8.5 – SSP5 | Represents a future in which high-end climate change occurs, while substantial emphasis is placed on social and economic development, fossil fuel exploitation and technology. | Increases in all capitals allow consistent surpluses of food and timber. Despite a slight general trend towards extensification, most of Europe remains under intensive management. | Very similar results across all parameterisations. |

**Table 2: Identities and characteristics of Representative Concentration Pathway – Shared Socioeconomic Pathway combinations used in CRAFTY-EU simulations presented here. Graphical results are shown in Supplement 2, and full descriptions of the scenarios used can be found in (Kok et al., 2018).**



**Appendix A: Model parameterisation**

This appendix describes the parameterisation of CRAFTY-EU, including the derivation of Agent Functional Types (AFTs). As outlined in the main text, AFT identities were designed to capture important sectoral and cross-sectoral land systems at local (10') scale. The initial distribution of these AFTs across the modelled land surface was then determined on the basis of

land use categories modelled by the IMPRESSIONS Integrated Assessment Platform (IAP) under baseline conditions (Harrison et al. 2015) (Fig. A1). This distribution ensured a common starting point for the two models that was fully consistent with the capital levels, demand levels and scenario conditions applied here, so that subsequent simulated changes could be attributed to changes in those conditions rather than inconsistencies in calibration data. The mapping of IAP output land use categories to AFTs is described in Table A1.

**Table A1: The composition of Agent Functional Types (AFTs) in *CRAFTY-EU* in terms of baseline IAP land use categories. In any case where the given IAP categories occupy more than 70% of a cell, that cell is allocated to the corresponding AFT in the baseline map of *CRAFTY-EU*, except in the case of the Peri-urban AFT, for which the threshold (of urban area) is 40%.**

| Agent Functional Type | Composition |
|---|---|
| Intensive arable farming | Intensively farmed |
| Intensive pastoral farming | Intensively grass |
| Intensive agro-forestry mosaic | Intensively farmed, intensively grass, managed forest |
| Intensive farming | Intensively farmed, intensively grass |
| Managed forestry | Managed forest |
| Mixed farming | Intensively farmed, intensively grass, extensively grass |
| Mixed pastoral farming | intensively grass, extensively grass, very extensively grass |
| Mixed forest | Managed forest, unmanaged forest |
| Extensive pastoral farming | Extensively grass |
| Extensive agro-forestry mosaic | extensively grass, very extensively grass, managed forest |
| Very extensive pastoral farming | Very extensively grass |
| Multifunctional | 4 or more land uses in uncommon combination |
| Minimal management | very extensively grass, unmanaged forest, unmanaged land |
| Unmanaged land | Unmanaged land |
| Unmanaged forest | Unmanaged forest |
| Peri-urban | Any combination with > 40% urban area |
| Urban | Urban |

The abilities of these AFTs to utilise capitals and produce ecosystem services were defined via capital sensitivity and

productive ability parameters (given, for each AFT, in Table A3). Where possible, values were derived from the IAP, and so preserved common forms of secondary land management and ecosystem service production within each AFT. Values that had no equivalent in the IAP (e.g. recreation service provision levels) were assumed on the basis of land management intensity and diversity, with variations used to understand the significance of these assumptions. This was also the case with the modelled biodiversity ecosystem service, which was here represented through the proxy of land use diversity (labelled Diversity below)

within each AFT.





In *CRAFTY-EU*, modelled production of ecosystem services occurs subject to capital levels, according to the equation

$$p_{s,i,t} = o_{s,t} \prod_c c_i{}^{\lambda_{c,t}} \tag{1}$$

Where p_(s,i,t) represents the level of production of ecosystem service s in cell i by AFT t, calculated as the product across all capitals c of cell-specific capital levels $c_i$ weighted by the sensitivity λ_(c,t) (black rows in tables below) of the AFT t to the capital c, multiplied by the maximum level of production o_(s,t) (red rows in tables below) that the AFT is able to produce.

Maximum production levels o_(s,t) and capital sensitivities λ_(c,t) are constant throughout simulations, while capital levels $c_i$ vary according to scenario and, potentially, previous production levels and institutional intervention. Maximum production levels can, however, vary across individual agents within AFTs, and do so here, in some experiments, randomly according to Gaussian distributions around the mean value (Tables A3 & A4).

The ability of an AFT to produce a service was first established by checking the average production level of each service across

cells assigned to that AFT under baseline conditions. If this average value was greater than or equal to 1% of the largest value produced by any AFT, that service was added to the AFT's productive abilities. The exact AFT-specific maximum production value (o_(s,t)) was calculated by extracting the 100 most productive cells for AFT t of service s and fitting a Gaussian distribution to the production levels in those cells using the R package fitdistrplus (Delignette-Muller and Dutang 2015). The mean of this fitted distribution was taken as the value of o_(s,t), while the standard deviation was retained for the introduction

of random variation in production levels. This procedure was used under the assumption that the 100 most productive cells represented optimal production conditions, and therefore provided a suitable basis to estimate production levels in the effective absence of capital constraints.

Capital levels were derived from outputs of the IAP to provide baseline and scenario-specific values (capitals are defined in Table A2). IAP results were interpolated to provide annual values for each capital on each grid cell within each scenario, for

the period 2010-2100. Where the derivation of capital values involves simulated quantities of production, these were normalised by the terrestrial area available in each cell (also an output of the IAP).

AFT-specific capital sensitivities λ_(c,t) were then estimated by plotting all production levels of service s by AFT t against each capital in turn (e.g. Fig. A1), with relationships quantified between the extremes of linear relationships (which were assigned a sensitivity value of 1.0) and random relationships (which were assigned a sensitivity value of 0.0). This procedure

did not, and was not intended to, replicate the land use allocation methods applied in the IAP, but to generate similar sensitivities on the basis of which agent decision-making could proceed.

Once these relationships were established, IAP output maps were used to quantify demand levels for each of the modelled ecosystem services by calculating service production levels according to the optimal production and capital sensitivity values described above. This was repeated at each timestep (2020s, 2050s and 2080s in the IAP, which were linearly interpolated to

annual values between 2016 and 2086 for CRAFTY-EU). Where the IAP projected a shortfall in service production, the supply was calculated and then scaled up to the equivalent of 100% to give a figure for demand. For the services not directly simulated by the IAP (recreation and diversity), the supply levels calculated from IAP output maps were taken as being equal to demand.



These demand levels (given in full in Appendix 3) were then used to calculate context-specific 'benefit' values of production as a basis for competition between agents. Benefit functions were defined to give the value of a certain level of production under a certain level of unmet demand, according to the equation:

$$m_s = u_s(r_s);$$

5     where $m_s$ is the marginal benefit for service s, $u_s$ is a function that describes the benefit (utility) of production of service s and $r_s$ is the residual demand for service s (Murray-Rust et al. 2014). Linear forms of $u_s$ were used here, calibrated to ensure equal relative valuation of services; i.e. the production of an equal proportion of unmet demand was assigned an equal benefit value whatever the service. This created a balanced competition between agents that was not skewed towards any particular service(s), with no benefit accruing from production when there was no unmet demand, prompting production under shortfalls

10     but not under surpluses.

**Table A2: Identities and details of modelled capitals. Exact parallels for some capitals were available in the IAP.**

| Capital | Explanation | Derivation from IAP |
|---|---|---|
| Crop productivity | Natural productivity for crops | Average of simulated productivities for winter wheat, spring wheat, winter barley, spring barley, potatoes, sugar beet, winter oilseed rape, spring oilseed rape, maize, forage maize, cotton, sunflower and soya |
| Grassland productivity | Natural productivity for grassland | Average of simulated productivities for grass, extensive grass and permanent grass |
| Forest productivity | Natural productivity for forest | Potential wood yield |
| Human capital | Availability of labour | Human capital |
| Social capital | General level of social support (cohesion, social networks) for production | Social capital |
| Manufactured capital | Availability of machinery and infrastructure (including for transportation of goods, where appropriate) | Manufactured capital |
| Financial capital | Economic resources supporting production | Financial capital |
| Urban capital | Suitability for urban development (used to constrain distribution of urban land to follow that modelled by the IAP) | Percentage urban cover of cell |





**Tables A3 a-q: Tables showing the sensitivities $\lambda_{c,t}$ of each AFT to capital levels and maximum service production levels $o_{s,t}$ (italics) (Eq. 1). Red values in brackets are the standard deviations of Gaussian distributions used in some simulations to randomly assign production levels to individual agents.**

**a) Intensive arable farming**

|              | Meat     | Crops      | Timber | Carbon   | Diversity   | Recreation  |
|--------------|----------|------------|--------|----------|-------------|-------------|
| Crop Prod    | 0        | 0.8        | 0      | 0        | 0           | 0           |
| Forest prod  | 0        | 0          | 0.7    | 0        | 0           | 0           |
| Grass Prod   | 1        | 0          | 0      | 0        | 0           | 0           |
| Financial    | 0.9      | 0.8        | 0.2    | 0        | 0           | 0.4         |
| Human        | 1        | 0.8        | 0.2    | 0        | 0           | 0.7         |
| Social       | 0.9      | 0.9        | 0.2    | 0        | 0           | 0.3         |
| Manufactured | 0.6      | 0.5        | 0.1    | 0        | 0           | 0.6         |
| Urban        | 0        | 0          | 0      | 0        | 0           | 0           |
| *Production* | *328 (40)* | *2280 (158)* | *0 (0)* | *422 (29)* | *0.54 (0.01)* | *0.1 (0.01)* |

**b) Intensive agro-forestry mosaic**

|              | Meat     | Crops     | Timber  | Carbon    | Diversity     | Recreation    |
|--------------|----------|-----------|---------|-----------|---------------|---------------|
| Crop Prod    | 0        | 0.3       | 0       | 0         | 0             | 0             |
| Forest prod  | 0.1      | 0.1       | 1       | 0         | 0             | 0             |
| Grass Prod   | 0.3      | 0         | 0       | 0         | 0.1           | 0             |
| Financial    | 0.6      | 0.7       | 0.2     | 0         | 0             | 0.4           |
| Human        | 0.5      | 0.8       | 0.1     | 0         | 0             | 0.7           |
| Social       | 0.5      | 0.6       | 0.3     | 0         | 0             | 0.3           |
| Manufactured | 0.2      | 0.2       | 0.2     | 0         | 0             | 0.6           |
| Urban        | 0        | 0         | 0       | 0         | 0             | 0             |
| *Production* | *316 (44)* | *811 (82)* | *59 (12)* | *481 (10)* | *0.082 (0.03)* | *0.15 (0.02)* |

**c) Intensive farming**

|              | Meat     | Crops      | Timber  | Carbon    | Diversity    | Recreation    |
|--------------|----------|------------|---------|-----------|--------------|---------------|
| Crop Prod    | 0.2      | 0.2        | 0.1     | 0         | 0            | 0             |
| Forest prod  | 0        | 0          | 0.6     | 0         | 0            | 0             |
| Grass Prod   | 0        | 0.1        | 0.1     | 0.2       | 0            | 0             |
| Financial    | 0.2      | 0.2        | 0.4     | 0.3       | 0            | 0.4           |
| Human        | 0.2      | 0.2        | 0.3     | 0.3       | 0            | 0.7           |
| Social       | 0.1      | 0.1        | 0.3     | 0.2       | 0            | 0.3           |
| Manufactured | 0.1      | 0          | 0.2     | 0         | 0            | 0.6           |
| Urban        | 0        | 0          | 0       | 0         | 0            | 0             |
| *Production* | *715 (75)* | *1064 (74)* | *15 (2)* | *466 (14)* | *0.75 (0.03)* | *0.15 (0.02)* |

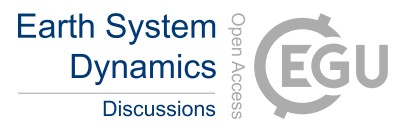

**d) Managed forest**

|  | Meat | Crops | Timber | Carbon | Diversity | Recreation |
|---|---|---|---|---|---|---|
| Crop Prod | 0 | 0.1 | 0 | 0 | 0 | 0 |
| Forest prod | 0.2 | 0 | 1 | 0.1 | 0 | 0 |
| Grass Prod | 0.3 | 0 | 0 | 0 | 0 | 0 |
| Financial | 0.3 | 0.1 | 0.2 | 0.1 | 0 | 0.4 |
| Human | 0.1 | 0.2 | 0.2 | 0.1 | 0 | 0.7 |
| Social | 0.2 | 0.2 | 0.2 | 0.1 | 0 | 0.3 |
| Manufactured | 0.1 | 0.1 | 0.3 | 0 | 0 | 0.6 |
| Urban | 0 | 0 | 0 | 0 | 0 | 0 |
| *Production* | *0 (0)* | *0 (0)* | *108 (18)* | *2412 (117)* | *0.51 (0.02)* | *0.5 (0.2)* |

**e) Extensive pastoral farming**

|  | Meat | Crops | Timber | Carbon | Diversity | Recreation |
|---|---|---|---|---|---|---|
| Crop Prod | 0.4 | 0 | 0 | 0 | 0 | 0 |
| Forest prod | 0 | 0 | 0.7 | 0 | 0 | 0 |
| Grass Prod | 0.2 | 0 | 0.4 | 0 | 0 | 0 |
| Financial | 0.1 | 0.1 | 0.2 | 0 | 0 | 0.4 |
| Human | 0.2 | 0 | 0.2 | 0 | 0 | 0.7 |
| Social | 0.2 | 0.1 | 0.1 | 0 | 0 | 0.3 |
| Manufactured | 0.2 | 0 | 0 | 0 | 0 | 0.6 |
| Urban | 0 | 0 | 0 | 0 | 0 | 0 |
| *Production* | *59 (45)* | *0 (0)* | *5 (6)* | *403 (59)* | *0.44 (0.04)* | *0.7 (0.1)* |

5  **f) Extensive agro-forestry mosaic**

|  | Meat | Crops | Timber | Carbon | Diversity | Recreation |
|---|---|---|---|---|---|---|
| Crop Prod | 0.1 | 0.3 | 0 | 0 | 0 | 0 |
| Forest prod | 0 | 0 | 1 | 0 | 0 | 0 |
| Grass Prod | 0.1 | 0.2 | 0.4 | 0 | 0 | 0 |
| Financial | 0.2 | 0.1 | 0.3 | 0 | 0 | 0.4 |
| Human | 0.2 | 0.1 | 0.3 | 0 | 0 | 0.7 |
| Social | 0.2 | 0.1 | 0.3 | 0 | 0 | 0.3 |
| Manufactured | 0.1 | 0 | 0.3 | 0 | 0 | 0.6 |
| Urban | 0 | 0 | 0 | 0 | 0 | 0 |
| *Production* | *105 (35)* | *0 (0)* | *57 (17)* | *634 (395)* | *0.72 (0.03)* | *0.7 (0.1)* |

**g) Multifunctional**

|  | Meat | Crops | Timber | Carbon | Diversity | Recreation |
|---|---|---|---|---|---|---|
| Crop Prod | 0 | 0.1 | 0 | 0 | 0 | 0 |
| Forest prod | 0 | 0 | 0.9 | 0 | 0 | 0 |
| Grass Prod | 0.7 | 0.1 | 0.1 | 0 | 0 | 0 |
| Financial | 0.7 | 0.3 | 0.2 | 0 | 0 | 0.4 |
| Human | 0.4 | 0.2 | 0.1 | 0 | 0 | 0.7 |
| Social | 0.5 | 0.3 | 0.1 | 0 | 0 | 0.3 |
| Manufactured | 0 | 0.2 | 0.2 | 0 | 0 | 0.6 |
| Urban | 0 | 0 | 0 | 0 | 0 | 0 |
| *Production* | *388 (50)* | *774 (132)* | *62 (12)* | *2232 (353)* | *0.89 (0.02)* | *0.5 (0.1)* |

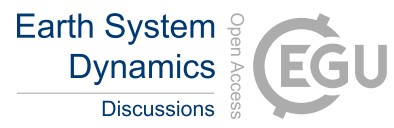

### h) Unmanaged forest

|               | Meat  | Crops | Timber | Carbon    | Diversity   | Recreation |
|---------------|-------|-------|--------|-----------|-------------|------------|
| Crop Prod     | 0     | 0     | 0      | 0         | 0           | 0          |
| Forest prod   | 0     | 0     | 0      | 0         | 0           | 0          |
| Grass Prod    | 0     | 0     | 0      | 0         | 0.3         | 0          |
| Financial     | 0     | 0     | 0      | 0         | 0           | 0.4        |
| Human         | 0     | 0     | 0      | 0         | 0           | 0.7        |
| Social        | 0     | 0     | 0      | 0         | 0           | 0.3        |
| Manufactured  | 0     | 0     | 0      | 0         | 0           | 0.6        |
| Urban         | 0     | 0     | 0      | 0         | 0           | 0          |
| *Production*  | *0 (0)* | *0 (0)* | *0 (0)* | *193 (94)* | *0.51 (0.02)* | *1 (0.1)* |

### i) Intensive pastoral farming

|               | Meat      | Crops | Timber | Carbon    | Diversity     | Recreation   |
|---------------|-----------|-------|--------|-----------|---------------|--------------|
| Crop Prod     | 0.2       | 0     | 0      | 0         | 0             | 0            |
| Forest prod   | 0         | 0     | 0      | 0         | 0             | 0            |
| Grass Prod    | 1         | 0     | 0      | 0         | 0             | 0            |
| Financial     | 0.8       | 0     | 0      | 0         | 0             | 0.4          |
| Human         | 0.6       | 0     | 0      | 0         | 0             | 0.7          |
| Social        | 0.7       | 0     | 0      | 0         | 0             | 0.3          |
| Manufactured  | 0.1       | 0     | 0      | 0         | 0             | 0.6          |
| Urban         | 0         | 0     | 0      | 0         | 0             | 0            |
| *Production*  | *799 (72)* | *0 (0)* | *0 (0)* | *513 (57)* | *0.51 (0.03)* | *0.1 (0.01)* |

5  ### j) Mixed farming

|               | Meat       | Crops       | Timber   | Carbon     | Diversity     | Recreation   |
|---------------|------------|-------------|----------|------------|---------------|--------------|
| Crop Prod     | 0.7        | 0.5         | 0        | 0          | 0             | 0            |
| Forest prod   | 0.1        | 0           | 0.8      | 0          | 0             | 0            |
| Grass Prod    | 0.2        | 0.2         | 0.1      | 0          | 0.2           | 0            |
| Financial     | 0.4        | 0.2         | 0.1      | 0          | 0             | 0.4          |
| Human         | 0.3        | 0.2         | 0.2      | 0          | 0             | 0.7          |
| Social        | 0.4        | 0.3         | 0.2      | 0          | 0             | 0.3          |
| Manufactured  | 0.2        | 0.1         | 0.2      | 0          | 0             | 0.6          |
| Urban         | 0          | 0           | 0        | 0          | 0             | 0            |
| *Production*  | *461 (54)* | *922 (132)* | *14 (4)* | *401 (35)* | *0.84 (0.03)* | *0.2 (0.02)* |

### k) Peri-urban

|               | Meat      | Crops       | Timber  | Carbon     | Diversity     | Recreation   |
|---------------|-----------|-------------|---------|------------|---------------|--------------|
| Crop Prod     | 0         | 0.3         | 0       | 0          | 0             | 0            |
| Forest prod   | 0.1       | 0           | 0.8     | 0          | 0             | 0            |
| Grass Prod    | 0.6       | 0.3         | 0       | 0          | 0             | 0            |
| Financial     | 0.5       | 0.1         | 0.4     | 0          | 0             | 0.4          |
| Human         | 0.4       | 0.1         | 0.2     | 0          | 0             | 0.7          |
| Social        | 0.3       | 0.1         | 0.2     | 0          | 0             | 0.3          |
| Manufactured  | 0.1       | 0.1         | 0.3     | 0          | 0             | 0.6          |
| Urban         | 1         | 1           | 1       | 1          | 1             | 1            |
| *Production*  | *86 (62)* | *143 (161)* | *9 (9)* | *404 (64)* | *0.64 (0.07)* | *0.2 (0.02)* |



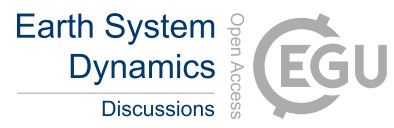

**l) Minimal management**

|  | Meat | Crops | Timber | Carbon | Diversity | Recreation |
|---|---|---|---|---|---|---|
| Crop Prod | 0 | 0 | 0 | 0 | 0 | 0 |
| Forest prod | 0 | 0 | 0 | 0 | 0 | 0 |
| Grass Prod | 0 | 0 | 0 | 0 | 0 | 0 |
| Financial | 0 | 0 | 0 | 0 | 0 | 0.4 |
| Human | 0 | 0 | 0 | 0 | 0.2 | 0.7 |
| Social | 0 | 0 | 0 | 0 | 0 | 0.3 |
| Manufactured | 0 | 0 | 0 | 0 | 0 | 0.6 |
| Urban | 0 | 0 | 0 | 0 | 0 | 0 |
| *Production* | *0 (0)* | *0 (0)* | *0 (0)* | *420 (13)* | *0.67 (0.03)* | *1 (0.1)* |

**m) Mixed pastoral**

|  | Meat | Crops | Timber | Carbon | Diversity | Recreation |
|---|---|---|---|---|---|---|
| Crop Prod | 0.2 | 0 | 0 | 0 | 0 | 0 |
| Forest prod | 0 | 0 | 0.7 | 0.1 | 0 | 0 |
| Grass Prod | 0.6 | 0 | 0.3 | 0 | 0 | 0 |
| Financial | 0.6 | 0 | 0.3 | 0 | 0 | 0.4 |
| Human | 0.6 | 0 | 0.2 | 0 | 0 | 0.7 |
| Social | 0.6 | 0 | 0.2 | 0 | 0 | 0.3 |
| Manufactured | 0.2 | 0 | 0.2 | 0 | 0 | 0.6 |
| Urban | 0 | 0 | 0 | 0 | 0 | 0 |
| *Production* | *484 (76)* | *0 (0)* | *9 (6)* | *491 (67)* | *0.7 (0.04)* | *0.35 (0.1)* |

**n) Unmanaged land**

|  | Meat | Crops | Timber | Carbon | Diversity | Recreation |
|---|---|---|---|---|---|---|
| Crop Prod | 0 | 0 | 0 | 0 | 0 | 0 |
| Forest prod | 0 | 0 | 0 | 0.7 | 0 | 0 |
| Grass Prod | 0 | 0 | 0 | 0 | 0 | 0 |
| Financial | 0 | 0 | 0 | 0 | 0 | 0.4 |
| Human | 0 | 0 | 0 | 0 | 0 | 0.7 |
| Social | 0 | 0 | 0 | 0 | 0 | 0.3 |
| Manufactured | 0 | 0 | 0 | 0 | 0 | 0.6 |
| Urban | 0 | 0 | 0 | 0 | 0 | 0 |
| *Production* | *0 (0)* | *0 (0)* | *0 (0)* | *2515 (254)* | *0.38 (0.05)* | *1 (0.1)* |

**o) Urban** (produces only urban area to replicate that simulated by the IAP)

|  | Meat | Crops | Timber | Carbon | Diversity | Recreation |
|---|---|---|---|---|---|---|
| Crop Prod | 0 | 0 | 0 | 0 | 0 | 0 |
| Forest prod | 0 | 0 | 0 | 0 | 0 | 0 |
| Grass Prod | 0 | 0 | 0 | 0 | 0 | 0 |
| Financial | 0 | 0 | 0 | 0 | 0 | 0 |
| Human | 0 | 0 | 0 | 0 | 0 | 0 |
| Social | 0 | 0 | 0 | 0 | 0 | 0 |
| Manufactured | 0 | 0 | 0 | 0 | 0 | 0 |
| Urban | 0 | 0 | 0 | 0 | 0 | 0 |
| *Production* | *0 (0)* | *0 (0)* | *0 (0)* | *0 (0)* | *0 (0)* | *0 (0)* |



**p) Mixed forest**

|             | Meat  | Crops | Timber   | Carbon   | Diversity   | Recreation |
|-------------|-------|-------|----------|----------|-------------|------------|
| Crop Prod   | 0     | 0     | 0        | 0        | 0           | 0          |
| Forest prod | 0     | 0     | 1        | 0        | 0           | 0          |
| Grass Prod  | 0     | 0     | 0        | 0        | 0           | 0          |
| Financial   | 0     | 0     | 0.3      | 0        | 0.2         | 0.4        |
| Human       | 0     | 0     | 0.4      | 0        | 0           | 0.7        |
| Social      | 0     | 0     | 0.1      | 0        | 0           | 0.3        |
| Manufactured| 0     | 0     | 0.1      | 0        | 0.1         | 0.6        |
| Urban       | 0     | 0     | 0        | 0        | 0           | 0          |
| *Production*| *0 (0)* | *0 (0)* | *61 (14)* | *356 (69)* | *0.51 (0.03)* | *1 (0.1)* |

**q) Very extensive pastoral**

|             | Meat  | Crops | Timber | Carbon | Diversity    | Recreation |
|-------------|-------|-------|--------|--------|--------------|------------|
| Crop Prod   | 0     | 0     | 0      | 0      | 0            | 0          |
| Forest prod | 0     | 0     | 0      | 0      | 0            | 0          |
| Grass Prod  | 0     | 0     | 0      | 0      | 0            | 0          |
| Financial   | 0     | 0     | 0      | 0      | 0.1          | 0.4        |
| Human       | 0     | 0     | 0      | 0      | 0.2          | 0.7        |
| Social      | 0     | 0     | 0      | 0      | 0.2          | 0.3        |
| Manufactured| 0     | 0     | 0      | 0      | 0.1          | 0.6        |
| Urban       | 0     | 0     | 0      | 0      | 0            | 0          |
| *Production*| *0 (0)* | *0 (0)* | *0 (0)* | *0 (0)* | *0.04 (0.03)* | *1 (0.1)* |





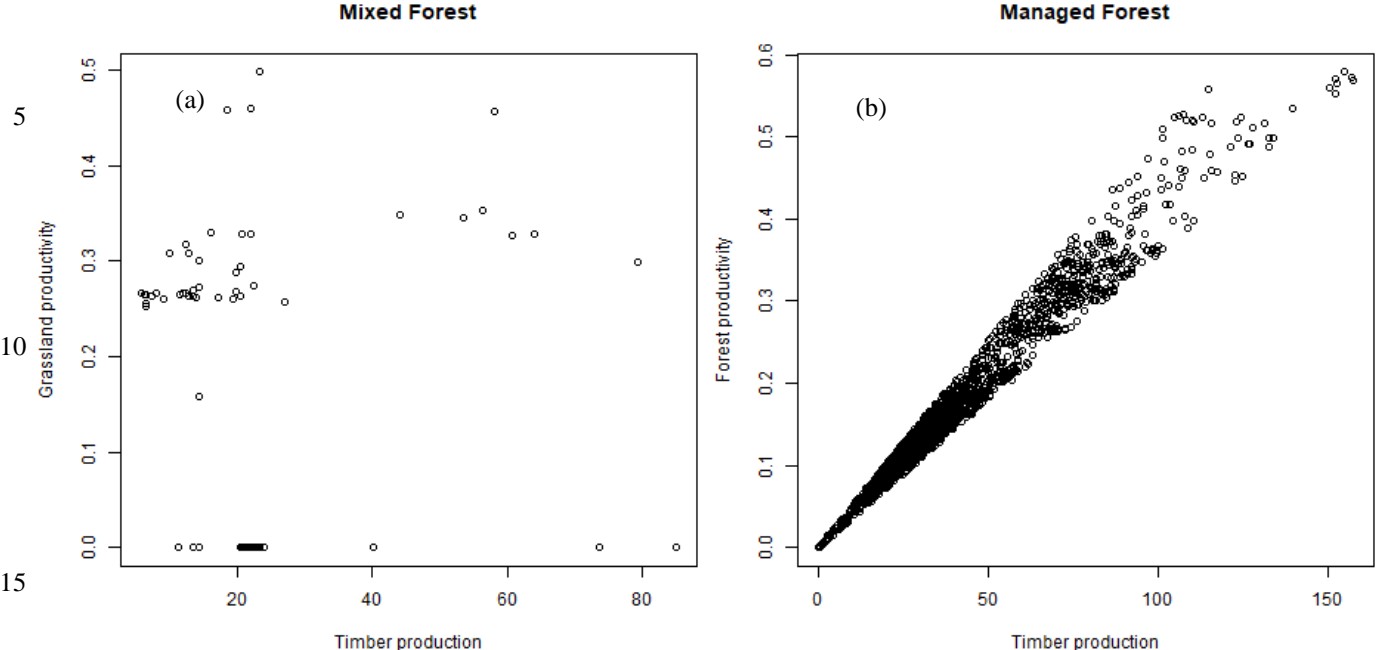

**Figure A1: Example capital-service relationships in IAP output data, used to quantify the capital sensitivities for AFTs in *CRAFTY-EU*. Timber production by Mixed Forest agents (a) is found to be almost completely insensitive to grassland productivity capital, giving a $\lambda_{c,t}$ value of 0, while timber production by Managed Forest agents (b) is highly sensitive to forest productivity, giving a $\lambda_{c,t}$**
20   **value of 1.**





**Table A4: Behavioural parameter variations used in the simulations. Parameter set 1 is the default from which main results are derived; in this setup agents respond directly to benefit values with no additional individual or typological behaviour. In parameter set 2, giving-up and giving-in thresholds are altered to introduce abandonment of land when benefit values fall below the giving-up threshold value, and resistance to change unless a competing land use has an additional benefit value of at least the giving-in threshold. Intensive land use agents are parameterised to be less tolerant of low benefit values, and more willing to switch to a land use with higher benefit values. In parameter set 3, individual agents differ from one another in terms of their abilities to produce different ecosystem services, and their giving-up and giving-in thresholds. Parameter sets 4 and 5 replicate parameter sets 2 and 3 respectively, but with larger values for thresholds and variations. Abbreviated AFT names are as follows: EP = Extensive Pastoral, Ext_AF = Extensive agro-forestry, IA = Intensive arable, Int_AF = Intensive agro-forestry, Int_Fa = Intensive farming, IP = Intensive pastoral, MF = Managed forest, Min_man = Minimal management, Mix_Fa = Mixed farming, Mix_For = Mixed forest, Mix_P = Mixed pastoral, Multifun = multifunctional, P-Ur = Peri-urban, UL = Unmanaged land, UMF = Unmanaged forest, Ur = Urban, VEP = Very extensive pastoral.**

**Param-set 1 (Behavioural baseline)**

| Name | givingInDistribution Mean | givingInDistribution nSD | givingUpDistribution Mean | givingUpDistribution nSD | serviceLevelNoise Min | serviceLevelNoise Max | givingUpProb |
|---|---|---|---|---|---|---|---|
| EP | 0.00000 | 0.00000 | 0.00000 | 0.00000 | 1.00000 | 1.00000 | 0.00000 |
| Ext_AF | 0.00000 | 0.00000 | 0.00000 | 0.00000 | 1.00000 | 1.00000 | 0.00000 |
| IA | 0.00000 | 0.00000 | 0.00000 | 0.00000 | 1.00000 | 1.00000 | 0.00000 |
| Int_AF | 0.00000 | 0.00000 | 0.00000 | 0.00000 | 1.00000 | 1.00000 | 0.00000 |
| Int_Fa | 0.00000 | 0.00000 | 0.00000 | 0.00000 | 1.00000 | 1.00000 | 0.00000 |
| IP | 0.00000 | 0.00000 | 0.00000 | 0.00000 | 1.00000 | 1.00000 | 0.00000 |
| MF | 0.00000 | 0.00000 | 0.00000 | 0.00000 | 1.00000 | 1.00000 | 0.00000 |
| Min_man | 0.00000 | 0.00000 | 0.00000 | 0.00000 | 1.00000 | 1.00000 | 0.00000 |
| Mix_Fa | 0.00000 | 0.00000 | 0.00000 | 0.00000 | 1.00000 | 1.00000 | 0.00000 |
| Mix_For | 0.00000 | 0.00000 | 0.00000 | 0.00000 | 1.00000 | 1.00000 | 0.00000 |
| Mix_P | 0.00000 | 0.00000 | 0.00000 | 0.00000 | 1.00000 | 1.00000 | 0.00000 |
| Multifun | 0.00000 | 0.00000 | 0.00000 | 0.00000 | 1.00000 | 1.00000 | 0.00000 |
| P-Ur | 0.00000 | 0.00000 | 0.00000 | 0.00000 | 1.00000 | 1.00000 | 0.00000 |
| UL | 0.00000 | 0.00000 | 0.00000 | 0.00000 | 1.00000 | 1.00000 | 0.00000 |
| UMF | 0.00000 | 0.00000 | 0.00000 | 0.00000 | 1.00000 | 1.00000 | 0.00000 |
| Ur | 0.00000 | 0.00000 | 100.00000 | 0.00000 | 1.00000 | 1.00000 | 0.00000 |
| VEP | 0.00000 | 0.00000 | 0.00000 | 0.00000 | 1.00000 | 1.00000 | 0.00000 |



**Param-set 2 (Increased thresholds)**

| Name | givingInDistribution Mean | givingInDistribution nSD | givingUpDistribution Mean | givingUpDistribution nSD | serviceLevelNoise Min | serviceLevelNoise Max | givingUpProb |
|---|---|---|---|---|---|---|---|
| EP | 0.00050 | 0.00000 | 0.00010 | 0.00000 | 1.00000 | 1.00000 | 0.10000 |
| Ext_AF | 0.00050 | 0.00000 | 0.00010 | 0.00000 | 1.00000 | 1.00000 | 0.10000 |
| IA | 0.00020 | 0.00000 | 0.00030 | 0.00000 | 1.00000 | 1.00000 | 0.25000 |
| Int_AF | 0.00020 | 0.00000 | 0.00030 | 0.00000 | 1.00000 | 1.00000 | 0.25000 |
| Int_Fa | 0.00020 | 0.00000 | 0.00030 | 0.00000 | 1.00000 | 1.00000 | 0.25000 |
| IP | 0.00020 | 0.00000 | 0.00030 | 0.00000 | 1.00000 | 1.00000 | 0.25000 |
| MF | 0.00020 | 0.00000 | 0.00010 | 0.00000 | 1.00000 | 1.00000 | 0.10000 |
| Min_man | 0.00050 | 0.00000 | 0.00010 | 0.00000 | 1.00000 | 1.00000 | 0.10000 |
| Mix_Fa | 0.00020 | 0.00000 | 0.00030 | 0.00000 | 1.00000 | 1.00000 | 0.25000 |
| Mix_For | 0.00050 | 0.00000 | 0.00010 | 0.00000 | 1.00000 | 1.00000 | 0.10000 |
| Mix_P | 0.00050 | 0.00000 | 0.00010 | 0.00000 | 1.00000 | 1.00000 | 0.10000 |
| Multifun | 0.00050 | 0.00000 | 0.00010 | 0.00000 | 1.00000 | 1.00000 | 0.10000 |
| P-Ur | 0.00050 | 0.00000 | 0.00010 | 0.00000 | 1.00000 | 1.00000 | 0.10000 |
| UL | 0.00050 | 0.00000 | 0.00010 | 0.00000 | 1.00000 | 1.00000 | 0.10000 |
| UMF | 0.00050 | 0.00000 | 0.00010 | 0.00000 | 1.00000 | 1.00000 | 0.10000 |
| Ur | 0.00050 | 0.00000 | 100.00000 | 0.00000 | 1.00000 | 1.00000 | 0.10000 |
| VEP | 0.00050 | 0.00000 | 0.00010 | 0.00000 | 1.00000 | 1.00000 | 0.10000 |





**Param-set 3 (Individual variation)**

| Name | givingInDistribution Mean | givingInDistribution nSD | givingUpDistribution Mean | givingUpDistribution nSD | serviceLevelNoise Min | serviceLevelNoise Max | givingUpProb |
|---|---|---|---|---|---|---|---|
| EP | 0.00050 | 0.00001 | 0.00010 | 0.00001 | 0.90000 | 1.10000 | 0.10000 |
| Ext_AF | 0.00050 | 0.00001 | 0.00010 | 0.00001 | 0.90000 | 1.10000 | 0.10000 |
| IA | 0.00020 | 0.00001 | 0.00030 | 0.00002 | 0.95000 | 1.05000 | 0.25000 |
| Int_AF | 0.00020 | 0.00001 | 0.00030 | 0.00002 | 0.95000 | 1.05000 | 0.25000 |
| Int_Fa | 0.00020 | 0.00001 | 0.00030 | 0.00002 | 0.95000 | 1.05000 | 0.25000 |
| IP | 0.00020 | 0.00001 | 0.00030 | 0.00002 | 0.95000 | 1.05000 | 0.25000 |
| MF | 0.00020 | 0.00001 | 0.00010 | 0.00001 | 0.95000 | 1.05000 | 0.25000 |
| Min_man | 0.00050 | 0.00001 | 0.00010 | 0.00001 | 0.90000 | 1.10000 | 0.10000 |
| Mix_Fa | 0.00020 | 0.00001 | 0.00030 | 0.00002 | 0.95000 | 1.05000 | 0.25000 |
| Mix_For | 0.00050 | 0.00001 | 0.00010 | 0.00001 | 0.90000 | 1.10000 | 0.10000 |
| Mix_P | 0.00050 | 0.00001 | 0.00010 | 0.00001 | 0.90000 | 1.10000 | 0.10000 |
| Multifun | 0.00050 | 0.00001 | 0.00010 | 0.00001 | 0.90000 | 1.10000 | 0.10000 |
| P-Ur | 0.00050 | 0.00001 | 0.00010 | 0.00001 | 1.00000 | 1.00000 | 0.10000 |
| UL | 0.00050 | 0.00001 | 0.00010 | 0.00001 | 0.90000 | 1.10000 | 0.10000 |
| UMF | 0.00050 | 0.00001 | 0.00010 | 0.00001 | 0.90000 | 1.10000 | 0.10000 |
| Ur | 0.00050 | 0.00001 | 100.00000 | 5.00000 | 1.00000 | 1.00000 | 0.10000 |
| VEP | 0.00050 | 0.00001 | 0.00010 | 0.00001 | 0.90000 | 1.10000 | 0.10000 |





**Param-set 4 (Larger Thresholds)**

| Name | givingInDistribution Mean | givingInDistribution nSD | givingUpDistribution Mean | givingUpDistribution nSD | serviceLevelNoise Min | serviceLevelNoise Max | givingUpProb |
|---|---|---|---|---|---|---|---|
| EP | 0.00100 | 0.00000 | 0.00030 | 0.00000 | 1.00000 | 1.00000 | 0.10000 |
| Ext_AF | 0.00100 | 0.00000 | 0.00030 | 0.00000 | 1.00000 | 1.00000 | 0.10000 |
| IA | 0.00040 | 0.00000 | 0.00100 | 0.00000 | 1.00000 | 1.00000 | 0.25000 |
| Int_AF | 0.00040 | 0.00000 | 0.00100 | 0.00000 | 1.00000 | 1.00000 | 0.25000 |
| Int_Fa | 0.00040 | 0.00000 | 0.00100 | 0.00000 | 1.00000 | 1.00000 | 0.25000 |
| IP | 0.00040 | 0.00000 | 0.00100 | 0.00000 | 1.00000 | 1.00000 | 0.25000 |
| MF | 0.00040 | 0.00000 | 0.00100 | 0.00000 | 1.00000 | 1.00000 | 0.25000 |
| Min_man | 0.00100 | 0.00000 | 0.00030 | 0.00000 | 1.00000 | 1.00000 | 0.10000 |
| Mix_Fa | 0.00040 | 0.00000 | 0.00100 | 0.00000 | 1.00000 | 1.00000 | 0.25000 |
| Mix_For | 0.00100 | 0.00000 | 0.00030 | 0.00000 | 1.00000 | 1.00000 | 0.10000 |
| Mix_P | 0.00100 | 0.00000 | 0.00030 | 0.00000 | 1.00000 | 1.00000 | 0.10000 |
| Multifun | 0.00100 | 0.00000 | 0.00030 | 0.00000 | 1.00000 | 1.00000 | 0.10000 |
| P-Ur | 0.00100 | 0.00000 | 0.00030 | 0.00000 | 1.00000 | 1.00000 | 0.10000 |
| UL | 0.00100 | 0.00000 | 0.00030 | 0.00000 | 1.00000 | 1.00000 | 0.10000 |
| UMF | 0.00100 | 0.00000 | 0.00030 | 0.00000 | 1.00000 | 1.00000 | 0.10000 |
| Ur | 0.00100 | 0.00000 | 100.00000 | 0.00000 | 1.00000 | 1.00000 | 0.10000 |
| VEP | 0.00100 | 0.00000 | 0.00030 | 0.00000 | 1.00000 | 1.00000 | 0.10000 |



**Param-set 5 (Larger Variations)**

| Name | givingInDistribution Mean | givingInDistribution nSD | givingUpDistribution Mean | givingUpDistribution nSD | serviceLevelNoise Min | serviceLevelNoise Max | givingUpProb |
|---|---|---|---|---|---|---|---|
| EP | 0.00050 | 0.00010 | 0.00010 | 0.00010 | 0.85000 | 1.15000 | 0.10000 |
| Ext_AF | 0.00050 | 0.00010 | 0.00010 | 0.00010 | 0.85000 | 1.15000 | 0.10000 |
| IA | 0.00020 | 0.00010 | 0.00030 | 0.00020 | 0.90000 | 1.10000 | 0.25000 |
| Int_AF | 0.00020 | 0.00010 | 0.00030 | 0.00020 | 0.90000 | 1.10000 | 0.25000 |
| Int_Fa | 0.00020 | 0.00010 | 0.00030 | 0.00020 | 0.90000 | 1.10000 | 0.25000 |
| IP | 0.00020 | 0.00010 | 0.00030 | 0.00020 | 0.90000 | 1.10000 | 0.25000 |
| MF | 0.00020 | 0.00010 | 0.00010 | 0.00010 | 0.85000 | 1.15000 | 0.10000 |
| Min_man | 0.00050 | 0.00010 | 0.00030 | 0.00020 | 0.90000 | 1.10000 | 0.25000 |
| Mix_Fa | 0.00020 | 0.00010 | 0.00010 | 0.00010 | 0.85000 | 1.15000 | 0.10000 |
| Mix_For | 0.00050 | 0.00010 | 0.00010 | 0.00010 | 0.85000 | 1.15000 | 0.10000 |
| Mix_P | 0.00050 | 0.00010 | 0.00010 | 0.00010 | 0.85000 | 1.15000 | 0.10000 |
| Multifun | 0.00050 | 0.00010 | 0.00010 | 0.00010 | 0.85000 | 1.15000 | 0.10000 |
| P-Ur | 0.00050 | 0.00010 | 0.00010 | 0.00010 | 1.00000 | 1.00000 | 0.10000 |
| UL | 0.00050 | 0.00010 | 0.00010 | 0.00010 | 0.85000 | 1.15000 | 0.10000 |
| UMF | 0.00050 | 0.00010 | 0.00010 | 0.00010 | 0.85000 | 1.15000 | 0.10000 |
| Ur | 0.00050 | 0.00010 | 100.00000 | 0.00010 | 1.00000 | 1.00000 | 0.10000 |
| VEP | 0.00050 | 0.00010 | 0.00010 | 0.00010 | 0.85000 | 1.15000 | 0.10000 |



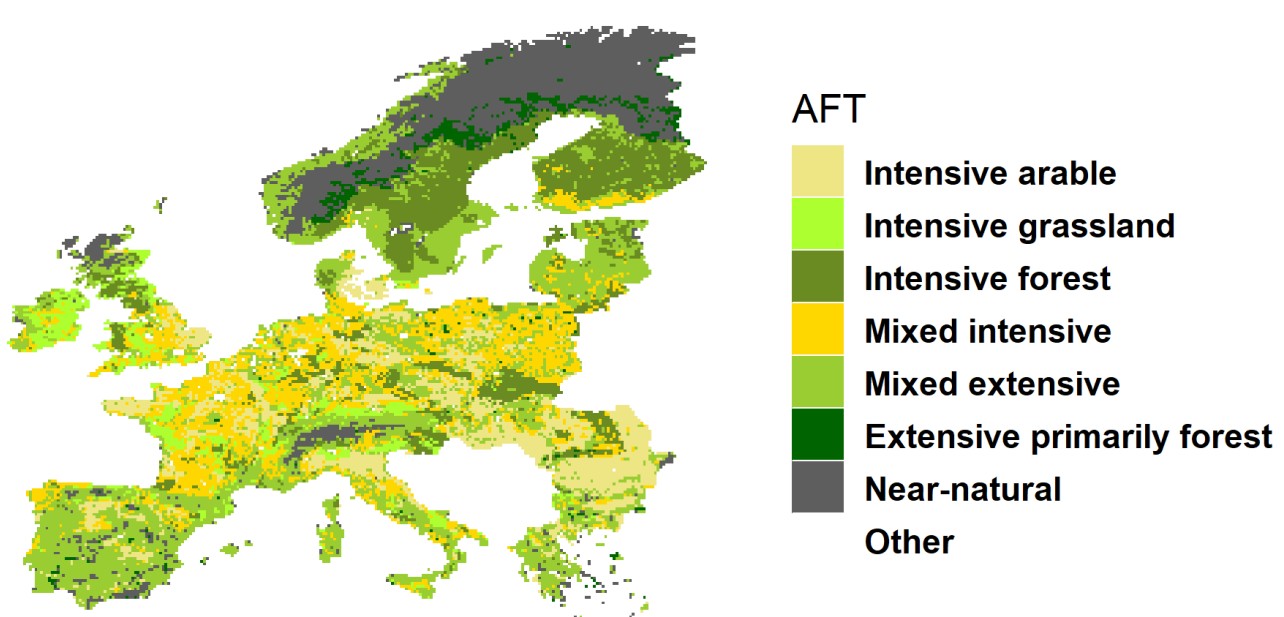

**Figure A2: Baseline *CRAFTY-EU* land cover from which all main simulations begin. This baseline map is derived from that of the IAP, which is a modelled land use allocation on the basis of 1961-1990 average climatic conditions and 2010 socio-economic conditions.**



## Appendix B: Model evaluation

Evaluation of CRAFTY-EU builds on previous evaluations of the agent-based modelling framework from which CRAFTY-EU is implemented, as well as evaluation of previous comparable implementations. These evaluations have included sensitivity and uncertainty analyses (Arneth, Brown and Rounsevell, 2014; Brown, Murray-Rust, et al., 2014; Murray-Rust et al., 2014a; Brown et al., 2016; Holzhauer, Brown and Rounsevell, 2018), model inter-comparison (Alexander et al., 2017; I. Holman et al., 2017), and validation against independently simulated and empirical data (V. Blanco, Brown, et al., 2017; V. Blanco, Holzhauer, et al., 2017). These evaluations are wholly or partially relevant to CRAFTY-EU as they deal, at least in part, with the basic architecture and parameters of the modelling framework, which are shared between all applications. The model is also fully open access, with code and (ODD+) descriptions of previous versions published ((Murray-Rust et al., 2014b; V. Blanco, Holzhauer, et al., 2017; Holzhauer, Brown and Rounsevell, 2018)), and with CRAFTY-EU itself available in full (code base) or for immediate use (interactive mode) online (from https://bitbucket.org/geoslurg/crafty_cobra_impressions_kit/src/default/ and https://crafty.shinyapps.io/CRAFTY-EU/). Furthermore, input data has been independently verified and evaluated during the development of the IMPRESSIONS IAP, from which CRAFTY-EU is calibrated (Harrison et al., 2012; Brown, Brown, et al., 2014; Kebede et al., 2015; I. P. Holman et al., 2017).

Evaluation here, therefore, focuses on the specific European implementation of CRAFTY. As described in the main text, evaluation comprised two main exercises involving runs under static, baseline conditions, the first starting from an unassigned (empty) land use map and the second from the baseline land use map derived from the IAP. The purpose of these two exercises was, respectively 1) to check whether baseline conditions would generate a 'realistic' land use configuration purely on the basis of capital levels and AFT characteristics (i.e. in the absence of any spatial information about land management), and 2) to check for divergence in outcomes from a common starting point consistent with other scenario runs.

The first exercise was conducted ten times to check the magnitude of stochastic variation in model outputs, and was expected to produce more variable outcomes for two reasons. Firstly, a number of potential 'solutions' exist to the problem of producing given levels of ecosystem services from a given landscape, and while reality represents one of these, models unconstrained by initial land use maps should be able to produce – and potentially transition between - many others. This is particularly likely here given the dependencies of simulated land use decisions on several different factors (multiple capitals, demand levels, and competition between agents). Furthermore, CRAFTY is a non-optimising and stochastic modelling framework with path-dependencies in outcomes, allowing individual simulations to diverge where initial conditions are unstable, as is the case here. Nevertheless, the degree of conformance in general characteristics of these simulations illuminates an important aspect of model stability, as well as revealing the predictability of model responses to aspatial input conditions.

The second exercise was simpler to interpret, with large differences in land use between the start and end of the simulation taken to indicate model instability under static conditions. Systematic changes would suggest an inconsistency between CRAFTY-EU parameterisation and baseline conditions, and random change would suggest a more general instability. Either of these would also suggest an innate bias in model outputs with the potential to obscure the impacts of simulated scenarios. Model outputs were therefore assessed in terms of the number of agents within each AFT over time.

The first exercise was initially performed over 800 timesteps, with 20% of cells being randomly selected for potential change. This long timespan and high rate of competition were chosen to exaggerate model dynamics, ensuring that they could be easily assessed through model outputs. Plots of AFT numbers and service levels were checked visually and statistically for stationarity (using Box-Ljung tests for temporal autocorrelation; (Ljung and Box, 1978)). Once an appropriate simulation duration had been identified, 10 further independent simulations were run to this point using different random number generator seed values. The outputs of these simulations were then compared in terms of total numbers of agents within each AFT, total service production levels, the spatial consistency of aggregated AFT classes across the ten simulations, and the similarity of these spatial patterns to that in the independent baseline map (to check for spontaneous convergence, which would suggest a broadly 'realistic' response to initial conditions). The second exercise was performed over 100 timesteps, again with a 20% rate of cell selection. This exercise was designed to run in a sufficient number of replicates to identify and understand



any divergence from stationarity in terms of numbers of agent per AFT, with stationarity again checked for both visually and statistically, and further runs used only where non-stationarity was detected.

Evaluation against historical data were not performed due to the lack of comprehensive data describing capital levels, demand levels and land use maps, other than those produced by alternative models (e.g. (Fuchs et al., 2015)).

## 5 Results

### 1st evaluation exercise

The first evaluation exercise did not result in stationarity during the 800-timstep run period (as confirmed by Box-Ljung tests, in which most AFT timeseries had p-values < 0.01 throughout the simulation). This suggests a tendency for ongoing oscillations in agent numbers (and hence service levels). Nevertheless, after an initial period of rapid change, all AFT numbers remained broadly consistent over time, with remaining short-term and apparent long-term fluctuations being small in comparison to overall agent numbers (Fig. B1). A cut-off of 300 timesteps (equivalent to year 2300 in the simulations) was chosen for further analysis, as AFT numbers had achieved representative values by this point.

The numbers of agents belonging to each AFT at the 300th timestep of each of the ten replicate simulations was very similar (Fig. B2a), as were the service levels produced (Fig. B2b). Furthermore, the spatial consistency of aggregated AFT classes was high, and locations frequently agreed with those in the independent baseline land use map (Fig. B3). Aggregated land use classes were used here to check the assignment of land uses rather than specific agent types, which, being considerably more numerous and less discrete, speak to a different aspect of model behaviour (the balance between competitive and productive behaviours of different AFTs, rather than the appropriateness of ecosystem service production in particular locations under given demand levels).



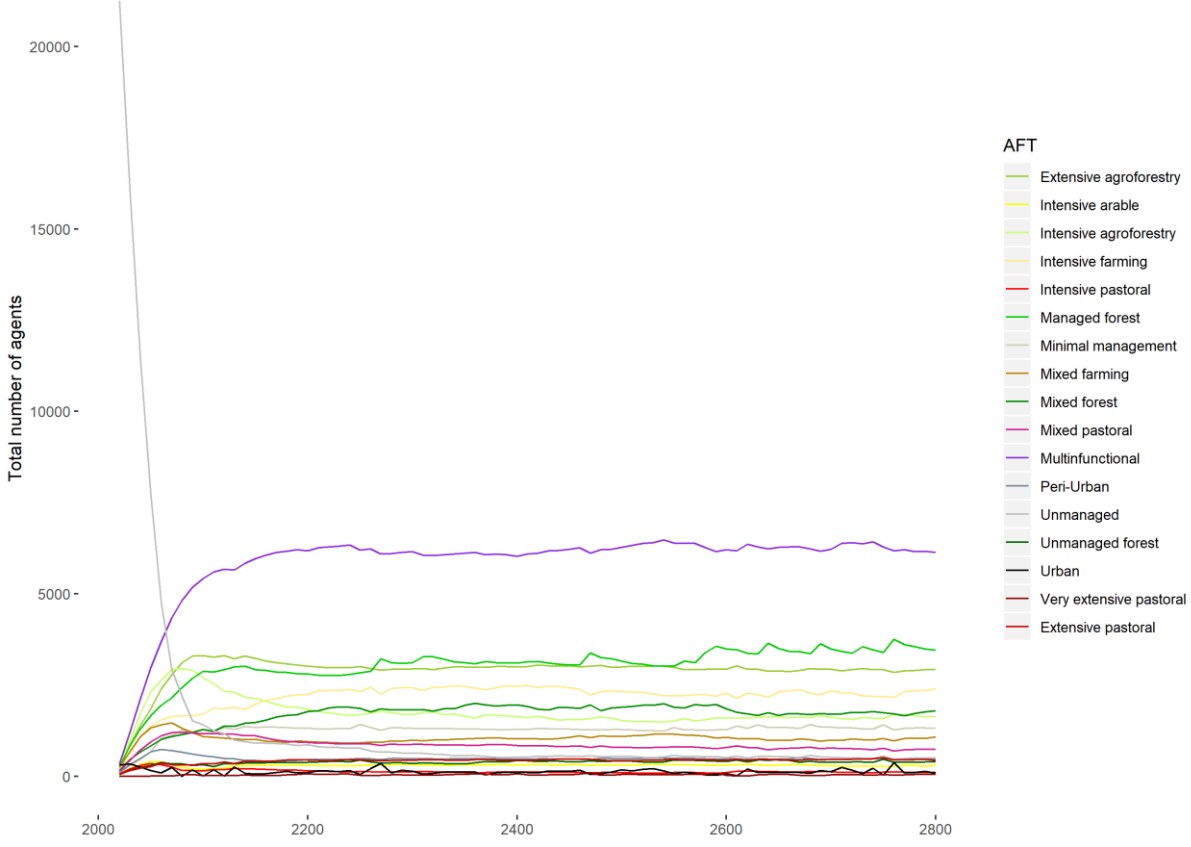

**Figure B1: Numbers of agents belonging to each Agent Functional Type throughout an 800-timestep simulation to check for stationarity.**



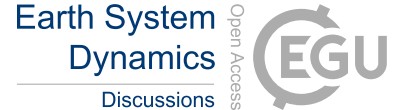

**Figure B2: Numbers of agents belonging to each Agent Functional Type at the 300th timestep of each of the ten independent simulations with no initial land use map (a), and service levels as a proportion of demand levels at the same points (b).**

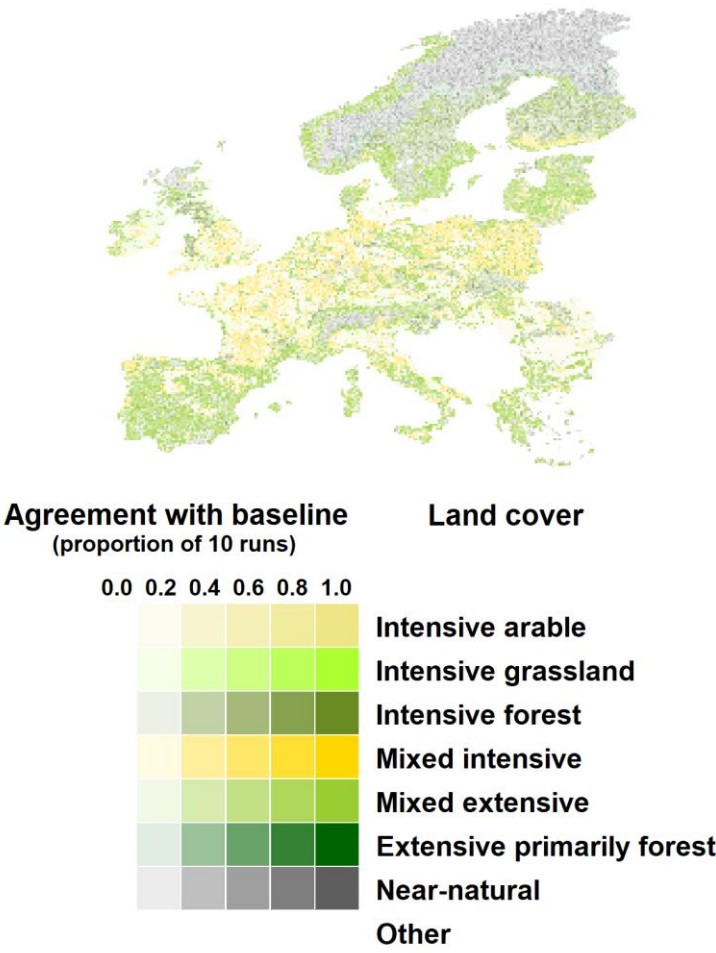

**Figure B3: Map of aggregated simulated land covers across the ten evaluation simulations initialised with no baseline land use map. Baseline land covers are shown on the map, with opacity scaled to show the number of evaluation simulations in which that land cover occurred at the 300th timestep.**



**2ⁿᵈ evaluation exercise**

The second evaluation exercise (running the model under baseline conditions starting from the baseline land use map) showed stationarity throughout the simulation period (Fig. B4), and this was confirmed by Box-Ljung tests that showed no evidence of dependence in the timeseries of any of the AFTs. Absolute numbers of agents remained within 15 of the initial number in all cases. This was taken to demonstrate stability in the initial configuration of *CRAFTY-EU*, implying that changes observed during scenario simulations were fully attributable to the parameterisation of those scenarios rather than inherent variability or trends in model dynamics.

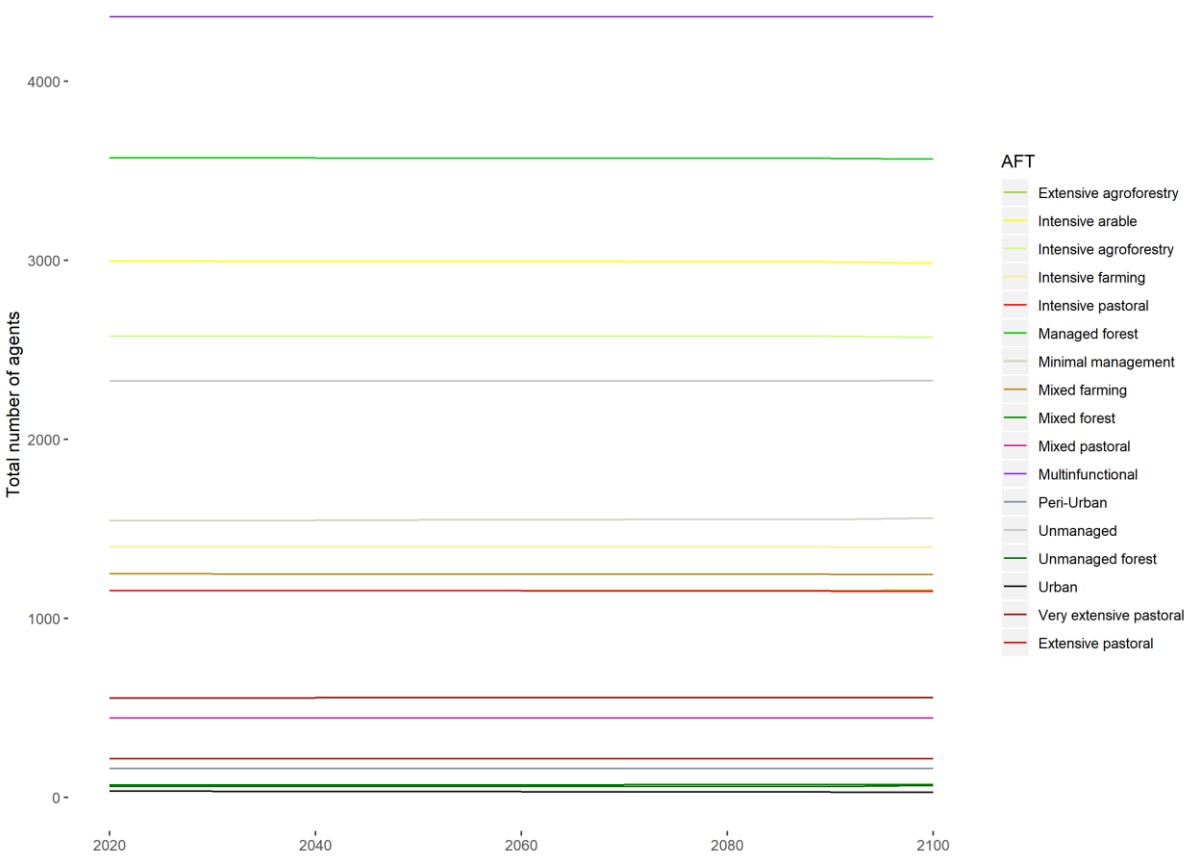

**Figure B4: Numbers of agents belonging to each Agent Functional Type throughout the 'baseline' run, in which *CRAFTY-EU* was initialised with the baseline land use map and run under static conditions.**