# Peer review of "Societal breakdown as an emergent property of large-scale behavioural models of land use change"

_Earth System Dynamics, 2019_

## Referee Comment (RC1) · Patrick Meyfroidt (Referee) · 20 Jun 2019

This manuscript presents a modelling experiment using a large-scale behavioral model of land-use change over Europe.

Overall, the research done is very valuable. In itself, it is not a breakthrough, as it builds on many earlier modelling efforts and explores one additional aspect of what can be done with this modelling framework, but it reveals a series of interesting insights.

However, in its present form I don't think that the manuscript is ready for publication. It misses details on some important methodological aspects as well as on some of the results that are at the core of the added value of the paper. These two issues are interlinked, in the sense that without these methodological details it is not really

possible to appreciate the value of some of the results, and these results themselves justify some of the methodological progresses.

So my main comment is to clarify key methodological aspects related to the ecosystem services demand and supply calculations, as well as the land use decision-making process, and to better present the results in particular related to the behavioral aspects. Most of my substantial comments are related to this. I return to this more in details below, and add two other substantial comments, on the evaluation of the scenarios based on "shortfall", and on the differences between standard economic models and this effort, as well as some minor ones.

1/ On gaps in methodological description and results presentation:

In general: I understand that the two modelling frameworks (Crafty and IAP) have been described elsewhere, but in order to understand the added value here more explanations are needed.

The behavioral part of the scenarios is very lightly, and unclearly, described. There's 3 lines on p.4 (l.24-26) to introduce the fact that the Agent Functional Types (AFT) have different behaviors, and then 10 lines p.7 (l.11-20) which describe very vaguely these different behavioral parameters. Beyond this, the basic behavioral and decision-making framework of the agents is not clearly described. The sentences on p.5, l.24-31 are very unclear to me. The description in the Appendix, in particular p.31, l.6-10 (which re-explains, but much more clearly, the p.5 l.29-31), and the caption of Table A4 p.38, are much more clear to me. This should be in the main text. Basically, we need to understand how agents make decision to either maintain or change land use, or how they are outcompeted by others. Just as two examples, in the present version it is impossible to understand how an agent can be "outcompeted by other agents" (p.7 l.16) or why "the more extreme behaviours [are] being selected out by a competitive process" (p.13, l.29).

This is similar when it comes to the results of this exploration. The results are introduced on p.9, l.25, but this is in fact referring to the Supplementary Material, and to Table 2 which is just a narrative summary of the results of the different scenarios, including the sensitivity of the behavioral parameters. But no actual result is presented in the main text. To me, this is insufficient.

As this is part of the title of the paper and one key aspect in the paper is to argue that such behavioral models are important to explore potential decision-makings that differ from monetary optimization, this would deserve more details. I do understand that one key conclusion is that is is not the behavioral parameters in themselves that matter so much, but rather their basic existence in the model, so that outcomes do not differ so much depending on the behavioral parameters but do differ between this model and others based on neoclassic economics. But still, if you want the reader to buy that idea, you really need to explain much more clearly how do the agents in this model, behaviorally-speaking, differ from basic monetary-optimizing agents that are implicit in many other LU models. And you should find a way to present, in the main paper, some of the results of the behavioral exploration. Currently, this is a set of graphs in Appendix 2 which will be, in essence, totally inexistent for most readers. I understand that this si lot of graphs to summarize in perhaps one or two Figures in the main text, and it probably requires some creativity, but I really think that discussing these results without presenting any of them in the main article is not correct.

The discussion on this (e.g., p.11 l.5-7) is itself very thin, and sometimes not very clear (e.g. p.13, l.5-11)..

2/ In addition, I have two other substantial comments:

2.1/ On the evaluation of the scenarios based on "shortfall": One key outcome on which the scenarios are discussed is the relative shortfall between demand and supply. This notion brings ambiguities. As discussed by the authors themselves (p.12), if supply really crashes because of socio-economic collapse, then at some point the demand will fall too, in a way that is not captured in the model. OK, but my concern also goes in the

other direction: The scenarios that correspond to high socio-economic development most likely generate a higher demand. If there is a shortfall in these higher demands, does that really mean that society's well-being is harmed? Is it possible to consider that some shortfall in a high-demand society reflects something more like a reasonable supply, which may lead to sufficient consumption? To formulate this in a less normative and more technical way, is it appropriate to only evaluate the outcomes in terms of the shortfall between demand and supply, or would it be reasonable to also evaluate the outcomes in terms of the overall (absolute value) of the supply?

2.2/ On the differences between standard economic models and this effort: The discussion, p.12 l.9-12, suggests that these economic models would be unable to represent such a collapse. I'm not totally clear about all the reasoning here. These models would indeed (l.10-12) display rising food prices, and thereby some maintenance of food production, but I'm not totally clear on how this would be so different than the results presented here - noting that the demand isn't adjusted here, as acknowledged by the authors. p.8, they say: "Conversely, where these capitals declined substantially, widespread extensification and abandonment of land occurred...": Yes, that makes sense, but you would expect this to also occur in standard economic models. What is precisely the argument?: - That this model predicts a much stronger decline in food production than standard economic models, then this has to be substantiated by numbers, - Or that this decline is more realistic than the lower decline in standard economic models, then this has to be justified convincingly. The idea in standard economic models that with production shortfalls, prices would rise, which would thus somehow buffer the production shortfall by mobilizing more capital towards agriculture is reasonable, especially considering that at some point anyone would have to admit that food is a basic need.

3/ A few more minor comments:

* Abstract: "economic irrationality": This is an ambiguous formulation. If one sticks to monetary profit optimization (I agree that this can be called "standard" economic models, but this needs to be explicit), lots of behaviors are irrational, if an "enlightened" economist expands a utility function to encompass pretty much anything, then it is hard to find any irrational behavior, and so on. (without entering into the whole discussion, things like imitation, sticking to one's behavior, and so on, can be perfectly rational under a given set of information and agentic capabilities). Better rephrase without such a connotation, or perhaps at least talk about irrationality in regards to monetary profit maximization. Next sentence, "this theoretical optimum" bears the same unclear connotations to me. This notion of "irrational" agent comes back later on and is misleading to me.

* p.2: "... where they are most required; when socio-ecological processes break down...": Yes, but this is only one example, any other situation of regime shift / systemic change / land-use transition brings similar challenges for basic land system models, be they based on economic rules or on statistical calibration.

* p.8: "..., which were not substantially reduced...": who is this "which"? The following sentence seems to suggest that you refer to the divergences in land system outcomes, but the sentence is odd as it is not correct to write that "divergent land systems (...) were not substantially reduced".

* p.9: l.1-15: This is described qualitatively. It would be good to find a way to present quantitatively the differences between these scenarios, in a way that would convince the reader of some of the points made, for example that SSP3 has such an impact compared to the climate scenario.

* p.11: l.14-17: Maybe yes, maybe no. This depends on how is the actual balance between ES, compared to your own way to balance them. But still I agree with the conclusion of the following lines that better understanding how these trade-offs are actually formulated in reality is crucial.

* p.11: l.23-24: and this would likely reinforce the shortfalls, right?

---

## Referee Comment (RC2) · Gunnar Dressler (Referee) · 3 Jul 2019

In this manuscript, the authors present an intriguing example of an agent-based model applied at the European scale that is able to produce realistic long-term projections of land-use change. The model – CRAFTY-EU – is an extension of the CRAFTY model that has been developed by the same author group. The main aim of the paper is therefore not to comprehensively present the model itself, but to rather to deliver a "proof-of-concept" that such an agent-based model a) can be applied at a large spatial scale and b) although it surely simplifies several relationships, e.g. regarding the representation of land-users, is well suited to explore long-term land-use change dynamics. It actually allows more flexibility in this regard, as it does not impose any constraints regarding optimality of the emerging land-use patterns and is therefore also able to simulate a –

although undesirable – breakdown of the land system.

Overall, the manuscript is well-written with regards to wording and grammar, and as such is pleasant to read. Especially the Introduction and Discussion Sections are clearly structured and the argumentation, backed by a range of references, is sound. However, the description of the model itself, and parts of the results, require some improvement, as they are not always fully and clearly described - which is also my main critique of the manuscript.

Parts of the manuscript – especially in the Methods $\&$ Results Sections – are a bit hard to follow, as the manuscript makes ample references to the Appendix here (11 references to the Appendix in Section 2, 6 references to the Appendix, respectively results in the Appendix in Section 3). At some points, it is therefore hard to follow the manuscript without jumping back and forth, respectively without fully reading the Appendix, in addition to the main text.

Examples in the **Methods** section are:

- p. 4, l. 24-26: "Behavioural differences between AFTs (in terms of willingness to change land use or abandon land, and range of variations in capital sensitivities and ecosystem service production levels) were also introduced to assess the robustness of model outcomes to behavioural variations (see below and Appendix A)" how these behavioural variations between AFTs is implemented is not clearly evident from the manuscript, there's only a very brief explanation on how behavioral parameters are varied at the end of Section 2.6 (p. 7, l. 11-14). As the authors stress that the representation of behavioral differences between AFTs is a crucial aspect of the model, some more information on this should be added to the main text.

- p. 5, l. 23-24: "with the agents producing the most (or the most highly valued) services gaining the highest benefit values and therefore best-placed to win the

competition for cells (Appendix A)" from this, it is not clear how agents compete for cells – in the appendix

In the **Results** Sections, this applies particularly to the model evaluation in Section 3.1, where the simulations starting with no initial land use map and with the baseline map are described. If these results are considered important, then the corresponding figures should also be included in the main text – otherwise, they should only be addressed in the Appendix.

**Questions regarding the Results:**

- Is there any explanation for the fact that "the model spontaneously produced realistic land use configurations on the basis of land productivities, AFT parameterisations and demand levels" (p. 8, l. 8-10) – i.e. the strength of external forcings?

- What are the main reasons for the "widespread extensification and abandonment of land occurred and large shortfalls in service levels" (p. 8, l. 24) – is it because agents only decide about profit?

- I did not fully understand the "peak effect" (p. 10, l. 1) with regard to individual behavioral variations and how irrational agents are selected out – can you explain this effect in more detail?

- Figure 1: Where do the sharp transitions in the mean capital value plots at specific points in time come from?

In the **Discussion** Section, the authors address both aspects regarding the model design (and its limitations), as well as regarding the model results. Although the authors clearly state that the current model represents a "a first-step towards improved understanding of behavioural processes within large-scale land systems." (p. 10, l. 30),

some of the simplifications that the model assumes could be addressed in more detail. One of the main limitations of the model is its relatively coarse resolution of only 10 arcminutes, and with that the limitation to only being able to represent generic agent functional types, rather than individual land managers. As such, the model is not able to represent individual land user's decisions or interactions. This does not diminish the results of the current study, but I would expect some statements here on a) how the focus on generic AFTs only vs. individual land managers may influence results and b) how a more detailed representation of individual land users may be achieved in future versions of the model.

**Questions regarding the Conclusions:**

• You state that "behavioural effects may be partially 'self-correcting', with more extreme behaviours being selected out by a competitive process." (p. 13, l. 28-29) – could show this, e.g. by plotting the distribution of behavioral parameters at the beginning and end of the simulation?

**Technical corrections:**

• p. 2, l. 30: "If a new generation of behavioural models are to make…" : are is

• p. 13, l. 18: "CRAFY-EU"

• Figure 1: please label the subfigures, e.g. a, b, c, d

• Figure 2: I found it very unintuitive to read the plot with negative values indicating a surplus, and positive values indicating a shortfall – could you show supply / demand instead, which would make it much easier to read over-/undersupply? Also, this would align better with Figure B2 in the appendix.

- Overall: in places, where references are cited with "e.g." , these references are enclosed in double brackets, e.g. p. 2, l. 23-35

**Appendix A & B:**

As the appendix includes a number of information that are quite essential to understand the manuscript, I would also suggest a number of minor corrections here:

1. Ordering of tables: You refer to Table A2 in the text only after referring to Tables A3 & A4 (p. 30, l. 19) – please correct the numbering.

2. Table A3: You refer to standard deviation values formatted in red, but table values are only black, this should be corrected. Also, the maximum service production levels are quite different from each other – what's the unit/scale of each service?

3. Table A4 that contains all the details on the variation of the behavioral parameters is only explained in the table caption – there's no explanation of the parameter variations in the text at all. This should definitely be added. Also, the Name column shows only abbreviations of the AFT names, which aren't used anywhere else – please use the full names here, as it the table is otherwise difficult to read and the column width is not limiting factor.

4. p. 31, l. 5-6: use subscript or _ for better readability for ms, us, rs, i.e. m_s, u_s, r_s

5. Figure B2 a) + b): readability of the figure would be improved if you add a grid to the figure, so that numbers of agents could be more easily compared, b) add a 1.0 line to make it easier to see whether an over-/undersupply occurs.

6. Figure B4: at first, I was wondering whether this figure made sense here, as numbers of agents per AFT did stay constant over the whole simulation run (as

per definition of the behavioral parameters in Table A4, where giving up / giving in thresholds are 0 in the baseline run) - only after careful examination I noticed some tiny changes towards the end of the simulation, e.g. for the Intensive Pastoral AFT - could you highlight these changes in the graph? Otherwise, I would omit this figure from the manuscript, as it does not provide a substantial benefit.

---

## Author Comment (AC1) · 10 Aug 2019

**Societal breakdown as an emergent property of large-scale behavioural models of land use change**

Calum Brown, Bumsuk Seo, and Mark Rounsevell

Responses to reviewers (our responses in red)

We're grateful to both reviewers for their detailed and insightful reviews, which raise a number of important points and provide helpful suggestions. We have revised the manuscript as described below.

**Reviewer 1 (Patrick Meyfroidt)**

This manuscript presents a modelling experiment using a large-scale behavioral model of land-use change over Europe. Overall, the research done is very valuable. In itself, it is not a breakthrough, as it builds on many earlier modelling efforts and explores one additional aspect of what can be done with this modelling framework, but it reveals a series of interesting insights.

Thanks for the positive comments.

However, in its present form I don't think that the manuscript is ready for publication. It misses details on some important methodological aspects as well as on some of the results that are at the core of the added value of the paper. These two issues are interlinked, in the sense that without these methodological details it is not really possible to appreciate the value of some of the results, and these results themselves justify some of the methodological progresses.

So my main comment is to clarify key methodological aspects related to the ecosystem services demand and supply calculations, as well as the land use decision-making process, and to better present the results in particular related to the behavioral aspects. Most of my substantial comments are related to this. I return to this more in details below, and add two other substantial comments, on the evaluation of the scenarios based on "shortfall", and on the differences between standard economic models and this effort, as well as some minor ones.

Many thanks for these comments. We fully accept the criticisms and are eager to ensure that the model is interpretable to readers, and have adopted all of the suggestions in the revised manuscript.

1/ On gaps in methodological description and results presentation:

In general: I understand that the two modelling frameworks (Crafty and IAP) have been described elsewhere, but in order to understand the added value here more explanations are needed.

The behavioral part of the scenarios is very lightly, and unclearly, described. There's 3 lines on p.4 (l.24-26) to introduce the fact that the Agent Functional Types (AFT) have different behaviors, and then 10 lines p.7 (l.11-20) which describe very vaguely these different behavioral parameters. Beyond this, the basic behavioral and decision-making framework of the agents is not clearly described. The sentences on p.5, l.24-31 are very unclear to me. The description in the Appendix, in particular p.31, l.6-10 (which re-explains, but much more clearly, the p.5 l.29-31), and the caption of Table A4 p.38, are much more clear to me. This should be in the main text. Basically, we need to understand how agents make decision to either maintain or change land use, or how they are outcompeted by others. Just as two examples, in the present version it is impossible to understand

how an agent can be "outcompeted by other agents" (p.7 l.16) or why "the more extreme behaviours [are] being selected out by a competitive process" (p.13, l.29).

We have now substantially extended (and tried to clarify) our presentation of the competition process and behavioural aspects of the model. We have included a new description of how these interact in the methods section (including by moving text from the Appendix into the main text as suggested), expanded the analysis of the different behavioural parameterisations in the results section, and given further interpretation of their effects in the discussion section (including the selection of behaviours that we previously suspected was occurring, and which we now show is a genuine but relatively weak, scenario-specific effect). We agree that understanding the operation of the model and the role of agent behaviours within it is necessary for interpreting the results, and hope our explanations are now sufficient.

This is similar when it comes to the results of this exploration. The results are introduced on p.9, l.25, but this is in fact referring to the Supplementary Material, and to Table 2 which is just a narrative summary of the results of the different scenarios, including the sensitivity of the behavioral parameters. But no actual result is presented in the main text. To me, this is insufficient.

As this is part of the title of the paper and one key aspect in the paper is to argue that such behavioral models are important to explore potential decision-makings that differ from monetary optimization, this would deserve more details. I do understand that one key conclusion is that is is not the behavioral parameters in themselves that matter so much, but rather their basic existence in the model, so that outcomes do not differ so much depending on the behavioral parameters but do differ between this model and others based on neoclassic economics. But still, if you want the reader to buy that idea, you really need to explain much more clearly how do the agents in this model, behaviorally-speaking, differ from basic monetary-optimizing agents that are implicit in many other LU models. And you should find a way to present, in the main paper, some of the results of the behavioral exploration. Currently, this is a set of graphs in Appendix 2 which will be, in essence, totally inexistent for most readers. I understand that this si lot of graphs to summarize in perhaps one or two Figures in the main text, and it probably requires some creativity, but I really think that discussing these results without presenting any of them in the main article is not correct.

The discussion on this (e.g., p.11 l.5-7) is itself very thin, and sometimes not very clear (e.g. p.13, l.5-11)..

We follow the general and specific suggestions here by:

1) Explaining more clearly the processes by which land use changes in the model (as for the above comment, but also specifically with respect to differences between this approach and purely economic or optimisation-based approaches) – methods section;
2) Further analysing the behavioural experiments and including new graphical and textual results of these in the main text and Appendix – results section, especially to show the effects of behavioural variations in the amended plot of supply-demand curves (Fig. 2) ;
3) Providing more interpretation of these effects in the discussion section;
4) Extending and clarifying the specific sections identified as being thin or unclear.

As noted in the comment, the behavioural variations are purely exploratory at this stage, and we do not wish to place undue weight on them or to overshadow our findings about the importance of 'supra-economic' behaviour in general, but hope that the revisions strike a better balance than did the original text.

2/ In addition, I have two other substantial comments:

2.1/ On the evaluation of the scenarios based on "shortfall": One key outcome on which the scenarios are discussed is the relative shortfall between demand and supply. This notion brings ambiguities. As discussed by the authors themselves (p.12), if supply really crashes because of socio-economic collapse, then at some point the demand will fall too, in a way that is not captured in the model. OK, but my concern also goes in the other direction: The scenarios that correspond to high socio-economic development most likely generate a higher demand. If there is a shortfall in these higher demands, does that really mean that society's well-being is harmed? Is it possible to consider that some shortfall in a high-demand society reflects something more like a reasonable supply, which may lead to sufficient consumption? To formulate this in a less normative and more technical way, is it appropriate to only evaluate the outcomes in terms of the shortfall between demand and supply, or would it be reasonable to also evaluate the outcomes in terms of the overall (absolute value) of the supply?

This is a very interesting point, thanks. We now interpret shortfalls and surpluses of food in terms of (approximate) per capita calorie consumption to account for the different demand levels between scenarios, and also discuss scenario implications in light of this point. We also feel that this highlights a general shortcoming in scenario modelling (including our own) in that the distribution of financial capital and hence access to food is rarely considered, and we include this point in the discussion.

2.2/ On the differences between standard economic models and this effort: The discussion, p.12 l.9-12, suggests that these economic models would be unable to represent such a collapse. I'm not totally clear about all the reasoning here. These models would indeed (l.10-12) display rising food prices, and thereby some maintenance of food production, but I'm not totally clear on how this would be so different than the results presented here - noting that the demand isn't adjusted here, as acknowledged by the authors. p.8, they say: "Conversely, where these capitals declined substantially, widespread extensification and abandonment of land occurred...": Yes, that makes sense, but you would expect this to also occur in standard economic models. What is precisely the argument?: - That this model predicts a much stronger decline in food production than standard economic models, then this has to be substantiated by numbers, - Or that this decline is more realistic than the lower decline in standard economic models, then this has to be justified convincingly. The idea in standard economic models that with production shortfalls, prices would rise, which would thus somehow buffer the production shortfall by mobilizing more capital towards agriculture is reasonable, especially considering that at some point anyone would have to admit that food is a basic need.

We agree that the reasoning here wasn't clear, and now discuss these issues in substantially more detail. The differences we now highlight between this model and the 'standard' economic paradigm are that:

1) A wider range of ecosystem services are included here, and the value of production of these services is more balanced;
2) These services, including food production, are sensitive to a wider range of capitals (including e.g. social and human capitals) – this is a major cause of the shortfalls we find in some scenarios, as economic inputs cannot entirely make up for the decline of other capitals;

3) In purely economic terms, both satisfactory prices and adequate financial capital are necessary to support modelled production (as calibrated in the AFT production functions), and financial capital here reflects the scenario conditions. Therefore, even if prices were entirely (and, we suggest, unrealistically) unconstrained, scenarios with low financial capital would be vulnerable to shortfalls in production;

4) Food prices here do rise as supply falls behind demand, but they do so according to a specified function that is not designed to guarantee the financial input necessary to achieve equilibrium – i.e. the prices are not artificially set at a level at which modelled shortfalls can be specified or eliminated;

5) The response of the model to price signals is ultimately determined by agent decision-making, which is not forced to be economically optimal – in many cases, small differences in the value of production do not stimulate changes in modelled land use;

6) Because no overall optimisation of land uses is used, the model is path-dependent and can move into states from which more productive or profitable alternatives are hard to reach (e.g. if a large number of marginal agents satisfy demand and/or are unwilling to change their land use, more efficient intensive agents may struggle to come in).

As the reviewer suggests, these differences are not necessarily fixed; many 'standard' economic models could certainly take a similar approach to most of these points. In the revised manuscript we therefore discuss these largely as differences in practice, rather than principle, and emphasise that the findings show the need for clear and well-supported assumptions regarding economic drivers and responses, rather than prioritising a particular approach *per se*.

3/ A few more minor comments:

* Abstract: "economic irrationality": This is an ambiguous formulation. If one sticks to monetary profit optimization (I agree that this can be called "standard" economic models, but this needs to be explicit), lots of behaviors are irrational, if an "enlightened" economist expands a utility function to encompass pretty much anything, then it is hard to find any irrational behavior, and so on. (without entering into the whole discussion, things like imitation, sticking to one's behavior, and so on, can be perfectly rational under a given set of information and agentic capabilities). Better rephrase without such a connotation, or perhaps at least talk about irrationality in regards to monetary profit maximization. Next sentence, "this theoretical optimum" bears the same unclear connotations to me. This notion of "irrational" agent comes back later on and is misleading to me.

Point taken, and we now avoid or explain the term 'irrationality'.

* p.2: "... where they are most required; when socio-ecological processes break down...": Yes, but this is only one example, any other situation of regime shift / systemic change / land-use transition brings similar challenges for basic land system models, be they based on economic rules or on statistical calibration.

This is true, and now acknowledged in the text.

* p.8: "..., which were not substantially reduced...": who is this "which"? The following sentence seems to suggest that you refer to the divergences in land system outcomes, but the sentence is odd as it is not correct to write that "divergent land systems (...) were not substantially reduced".

Corrected, thanks.

* p.9: l.1-15: This is described qualitatively. It would be good to find a way to present quantitatively the differences between these scenarios, in a way that would convince the reader of some of the points made, for example that SSP3 has such an impact compared to the climate scenario.

We now include some quantitative comparisons between the scenarios as suggested.

* p.11: l.14-17: Maybe yes, maybe no. This depends on how is the actual balance between ES, compared to your own way to balance them. But still I agree with the conclusion of the following lines that better understanding how these trade-offs are actually formulated in reality is crucial.

We have rephrased to emphasise the latter point.

* p.11: l.23-24: and this would likely reinforce the shortfalls, right?

Correct, added.

**Reviewer 2 (Gunnar Dressler)**

In this manuscript, the authors present an intriguing example of an agent-based model applied at the European scale that is able to produce realistic long-term projections of land-use change. The model – CRAFTY-EU – is an extension of the CRAFTY model that has been developed by the same author group. The main aim of the paper is therefore not to comprehensively present the model itself, but to rather to deliver a "proof-of-concept" that such an agent-based model a) can be applied at a large spatial scale and b) although it surely simplifies several relationships, e.g. regarding the representation of land-users, is well suited to explore long-term land-use change dynamics. It actually allows more flexibility in this regard, as it does not impose any constraints regarding optimality of the emerging land-use patterns and is therefore also able to simulate a – although undesirable – breakdown of the land system.

Overall, the manuscript is well-written with regards to wording and grammar, and as such is pleasant to read. Especially the Introduction and Discussion Sections are clearly structured and the argumentation, backed by a range of references, is sound. However, the description of the model itself, and parts of the results, require some improvement, as they are not always fully and clearly described - which is also my main critique of the manuscript.

Thanks for the positive comments and identifying these shortcomings (which we note largely agree with those identified by Patrick Meyfroidt).

Parts of the manuscript – especially in the Methods & Results Sections – are a bit hard to follow, as the manuscript makes ample references to the Appendix here (11 references to the Appendix in Section 2, 6 references to the Appendix, respectively results in the Appendix in Section 3). At some points, it is therefore hard to follow the manuscript without jumping back and forth, respectively without fully reading the Appendix, in addition to the main text.

Yes, we accept this point. We now include more detail from the Appendix directly in the main text (methods section especially), and have also expanded the description of how the model operates as outlined above. These revisions are intended to minimise the need for readers to refer to the Appendix in order to understand the results we present.

Examples in the Methods section are:

• p. 4, l. 24-26: "Behavioural differences between AFTs (in terms of willingness to change land use or abandon land, and range of variations in capital sensitivities and ecosystem service production levels) were also introduced to assess the robustness of model outcomes to behavioural variations (see below and Appendix A)" how these behavioural variations between AFTs is implemented is not clearly evident from the manuscript, there's only a very brief explanation on how behavioral parameters are varied at the end of Section 2.6 (p. 7, l. 11-14). As the authors stress that the representation of behavioral differences between AFTs is a crucial aspect of the model, some more information on this should be added to the main text.

We have now substantially revised and extended this section (we particularly focused on this as both reviewers made similar points here).

• p. 5, l. 23-24: "with the agents producing the most (or the most highly valued) services gaining the highest benefit values and therefore best-placed to win the competition for cells (Appendix A)" from this, it is not clear how agents compete for cells – in the appendix.

This has also now been expanded and clarified.

In the Results Sections, this applies particularly to the model evaluation in Section 3.1, where the simulations starting with no initial land use map and with the baseline map are described. If these results are considered important, then the corresponding figures should also be included in the main text – otherwise, they should only be addressed in the Appendix.

We are not entirely convinced of this – we believe it's important for readers to know that the model behaved in this way (achieving a reasonable outcome when initiated without a starting map, and remaining static when run under static conditions) because it speaks directly to the interpretation and reliability of the main results. However, we're not sure that the figures themselves merit inclusion in the main text, as they don't really reveal anything further. We therefore keep the existing structure for now, with the figures available in the Appendix for readers who wish to check the claims we make in the text about evaluation.

Questions regarding the Results:

• Is there any explanation for the fact that "the model spontaneously produced realistic land use configurations on the basis of land productivities, AFT parameterisations and demand levels" (p. 8, l. 8-10) – i.e. the strength of external forcings?

We now explain this result in slightly more detail in the text, to say that we would expect some level of agreement with reality because the model was given capital levels and demand levels, which are the major drivers of land use. However, we would not expect complete agreement because there are numerous potential configurations of land use that would represent 'reasonable' outcomes, and a model unconstrained by an initial map should be able to achieve many of these. While it is hard to be precise in the interpretation of this result, we suggest that it shows that the use of capitals and demands allows at least the possibility of reproducing observed land use dynamics – i.e. that the model responds to these basic drivers broadly as real land systems do.

• What are the main reasons for the "widespread extensification and abandonment of land occurred and large shortfalls in service levels" (p. 8, l. 24) – is it because agents only decide about profit?

This is also similar to comments above about the role of economic and non-economic behaviour in the model. As outlined there, we have substantially extended our discussion of these findings to explain these outcomes – which develop largely because agents are reliant on a range of capitals

(natural productivity, human, social, manufactured and financial capitals), several of which decrease dramatically in some scenarios. This leaves agents unable to achieve a level of service production that brings returns (economic & non-economic) defined as acceptable.

• I did not fully understand the "peak effect" (p. 10, l. 1) with regard to individual behavioral variations and how irrational agents are selected out – can you explain this effect in more detail?

This point was largely speculative in the initial manuscript, but in response to this suggestion we have looked into the evolution of behaviour in more detail. We found that behavioural parameters do systematically change in some scenarios, and describe and attempt to explain this in the text and in new figures in the Appendix. This (slight) effect seems to be due to the persistence of agents with tolerance of low benefit values and unwillingness to change land use in more challenging scenarios (particularly SSP3).

• Figure 1: Where do the sharp transitions in the mean capital value plots at specific points in time come from?

These come from the timesteps at which capital values were available from the IAP (between which values were linearly interpolated). We have added and explanation to the figure legend.

In the Discussion Section, the authors address both aspects regarding the model design (and its limitations), as well as regarding the model results. Although the authors clearly state that the current model represents a "a first-step towards improved understanding of behavioural processes within large-scale land systems." (p. 10, l. 30), some of the simplifications that the model assumes could be addressed in more detail. One of the main limitations of the model is its relatively coarse resolution of only 10 arcminutes, and with that the limitation to only being able to represent generic agent functional types, rather than individual land managers. As such, the model is not able to represent individual land user's decisions or interactions. This does not diminish the results of the current study, but I would expect some statements here on a) how the focus on generic AFTs only vs. individual land managers may influence results and b) how a more detailed representation of individual land users may be achieved in future versions of the model.

Thanks for the suggestion, and we agree that these are important considerations. We now discuss both of these points in the discussion section, in light of this application as well as basic CRAFTY design (based on the premise that an informative balance between important behaviour and 'noise' exists somewhere above the individual level – e.g. at the level of 'functional types'), as well as other models' findings and discussion in the literature.

Questions regarding the Conclusions:

• You state that "behavioural effects may be partially 'self-correcting', with more extreme behaviours being selected out by a competitive process." (p. 13, l. 28- 29) – could show this, e.g. by plotting the distribution of behavioral parameters at the beginning and end of the simulation?

Thanks for this suggestion. We have done this, and include the resulting plots in the Appendix with further discussion of the effect in the main text.

Technical corrections:

• p. 2, l. 30: "If a new generation of behavioural models are to make. . . " : are is

Corrected, thanks

• p. 13, l. 18: "CRAFY-EU"

Corrected, thanks

• Figure 1: please label the subfigures, e.g. a, b, c, d

Done

• Figure 2: I found it very unintuitive to read the plot with negative values indicating a surplus, and positive values indicating a shortfall – could you show supply / demand instead, which would make it much easier to read over-/undersupply? Also, this would align better with Figure B2 in the appendix.

Changed as suggested (also with the addition of ranges to show the effects of behavioural variations).

• Overall: in places, where references are cited with "e.g." , these references are enclosed in double brackets, e.g. p. 2, l. 23-35

Thanks, now corrected

Appendix A & B: As the appendix includes a number of information that are quite essential to understand the manuscript, I would also suggest a number of minor corrections here:

1. Ordering of tables: You refer to Table A2 in the text only after referring to Tables A3 & A4 (p. 30, l. 19) – please correct the numbering.

Thanks; this table should have been referred to earlier, and now is.

2. Table A3: You refer to standard deviation values formatted in red, but table values are only black, this should be corrected. Also, the maximum service production levels are quite different from each other – what's the unit/scale of each service?

We have corrected the tables to refer to the bracketed rather than red values, and to include the standardised production values that form the basis of the model calculations. The original production levels do have very different units and are a mix of empirically-based values (e.g. extractive goods) and abstract values (e.g. recreation).

3. Table A4 that contains all the details on the variation of the behavioral parameters is only explained in the table caption – there's no explanation of the parameter variations in the text at all. This should definitely be added. Also, the Name column shows only abbreviations of the AFT names, which aren't used anywhere else – please use the full names here, as it the table is otherwise difficult to read and the column width is not limiting factor.

We have made these changes, and added the paramerisation explanation to the main text, Section 2.6.

4. p. 31, l. 5-6: use subscript or _ for better readability for ms, us, rs, i.e. $m_s$, $u_s$, $r_s$

Thanks for spotting this, we have changed to subscripts.

5. Figure B2 a) + b): readability of the figure would be improved if you add a grid to the figure, so that numbers of agents could be more easily compared, b) add a 1.0 line to make it easier to see whether an over-/undersupply occurs.

Done, thanks for the suggestions

6. Figure B4: at first, I was wondering whether this figure made sense here, as numbers of agents per AFT did stay constant over the whole simulation run (as per definition of the behavioral parameters

in Table A4, where giving up / giving in thresholds are 0 in the baseline run) - only after careful examination I noticed some tiny changes towards the end of the simulation, e.g. for the Intensive Pastoral AFT - could you highlight these changes in the graph? Otherwise, I would omit this figure from the manuscript, as it does not provide a substantial benefit.

It's correct that the plot shows effectively no change (agent numbers are almost completely constant, as described in section 3.1.2), but we find this an important if not particularly dramatic result. The giving up and giving in thresholds do not preclude change here – while agents will tolerate any positive benefit (giving up threshold), they will also change to a land use with any advantage (giving in threshold) – i.e. they are actually highly sensitive to competitive advantages. As a result the plot demonstrates model stability under static conditions rather than a predetermined result of parameterisation, and we believe showing this visually in the appendix may be a useful addition to saying so in the main text.

---

## Author Response (AR2)

Thank you for the additional comments. Again we found these useful and have adopted almost all in the enclosed revision.

Section 2.4.: the description of agent competition for cells is still not entirely clear to me: You state that "agents did not optimise their land uses according to benefit values, and these values were not used to ensure full supply of each service. Instead benefit values responded in defined ways to changes in demand and supply levels, stimulating production, but not guaranteeing a given production level." (p.6., l. 1-5). However, in the baseline case where there are no differences in behavioral parameters, agents are fully rational profit maximizers – and although I understand that the behavioral thresholds of abandonment and competition define why agents may persist with a land use, although it is not the one with the highest benefit, I would rather understand this as a constrained optimization, where the thresholds define some boundary conditions. If this is not the case, you should state more clearly how "benefit values responded in defined ways to changes in demand and supply levels" – i.e. what these defined ways are.

This raises an important point, which we address in two ways: 1) explaining in new text in Section 2.4 that individual agents will indeed behave in a benefit-maximising way if thresholds are set to 0, although this does not lead to overall optimisation because it depends on the rate at which cells are subjected to competition (previously mentioned first in Section 2.6); 2) clarifying that benefit values are mathematically defined responses to demand and supply levels, upon which agent decision-making is based.

The parameter description in section 2.6. is much better than in the previous version. However, what I am missing is a general overview on the behavioral parameters and their meaning: you mention that AFTs differ in their behavior in section 2.2 (pointing to section 2.6). Then you mention the competition and abandonment threshold first in section 2.4. (without stating that these are the behavioral traits of the AFTs), and finally you describe how behavioral paramaters are varied in different sets in section 2.6. – without referring to what parameters are included in these sets, which only becomes apparent in the appendix. Without consulting the appendix, the reader has no clear idea about the behavioral parameters. Adding a short table to the main text with an overview on the parameters and a short definition would drastically improve the understanding – especially as the behavioral differentiation of the AFTs is such a central part of the manuscript.

We appreciate this suggestion and have added a table to the manuscript (Table 3).

Results section:

Some ambiguities still exist in the paragraph on behavioral parameter variation (section 3.2, p. 10, l.7-27): You state that "In two scenarios (RCP2.6-SSP4 and RCP4.5-SSP3), behavioural parameterisations determined whether food was over- or under-supplied by the 2020s" (p.10, l. 13-15) – but you don't state which behavioral parameter set led to which outcome.

We have now clarified this in the text.

I very much appreciate the further simulations and in detail investigation on the effects of the behavioral parameters and you state how the distribution of behavioral parameters changes - "higher values of competition thresholds, lower values of abandonment thresholds and lower variation between agents demonstrating a disproportionate persistence of agents who are relatively unlikely to respond to benefit values" (p. 10, l. 25-26) – but you don't state the reasons for the shift

towards these values. Couldn't there be also cases where mostly agent's that are always responding to benefit values persist?

Yes, we suspect this could occur, but did not find such an effect in these experiments. We have added this observation to the text.

In general, I still think that more of the analysis and explanation of the behavioral parameter variation should be placed in the main text, as it is a core part of the model and highlights the "added value" of the model, compared to other large scale land-use models.

We appreciate and understand the suggestion, but are concerned about the (already long and detailed) paper becoming too unwieldy. Currently we hope that the manuscript strikes a balance between highlighting the general effect of a behavioural model in moving away from equilibrium dynamics and illustrating the roles of particular behaviours within that effect. We certainly agree that those behaviours merit more attention, but feel that follow-up studies are probably required to explore and present them fully.

Figure 2: The figure is much better now but consider adding a grid to improve the readability (similar to Figure B2).

We have now done so

Appendix C:

Figure C1: You use fractal dimension as a measure – this is not a standard measure, so please shortly state, what it means.

We have now added a sentence in Section 2.6 to define this measure

[revised manuscript text omitted]